# LISA: Learning Interpretable Skill Abstractions from Language

**Divyansh Garg**[*]     **Skanda Vaidyanath**[*]     **Kuno Kim**
**Jiaming Song**     **Stefano Ermon**
Stanford University
{divgarg, svaidyan, khkim, tsong, ermon} @stanford.edu

## Abstract

Learning policies that effectively utilize language instructions in complex, multi-task environments is an important problem in sequential decision-making. While it is possible to condition on the entire language instruction directly, such an approach could suffer from generalization issues. In our work, we propose *Learning Interpretable Skill Abstractions (LISA)*, a hierarchical imitation learning framework that can learn diverse, interpretable *primitive behaviors* or skills from language-conditioned demonstrations to better generalize to unseen instructions. LISA uses vector quantization to learn discrete skill codes that are highly correlated with language instructions and the behavior of the learned policy. In navigation and robotic manipulation tasks, LISA outperforms a strong non-hierarchical Decision Transformer baseline in the low data regime and is able to compose learned skills to solve tasks containing unseen long-range instructions. Our method demonstrates a more natural way to condition on language in sequential decision-making problems and achieve interpretable and controllable behavior with the learned skills.

## 1 Introduction

Intelligent machines should be able to solve a variety of complex, long-horizon tasks in an environment and generalize to novel scenarios. In the sequential decision-making paradigm, provided expert demonstrations, an agent can learn to perform these tasks via multi-task imitation learning (IL). As humans, it is desirable to specify tasks to an agent using a convenient, yet expressive modality and the agent should solve the task by taking actions in the environment. There are several ways for humans to specify tasks to an agent, such as task IDs, goal images, and goal demonstrations. However, these specifications tend to be ambiguous, require significant human effort, and can be cumbersome to curate and provide at test time. One of the most natural and versatile ways for humans to specify tasks is via natural language.

The goal of language-conditioned IL is to solve tasks in an environment given language-conditioned trajectories at training time and a natural language instruction at test time. This becomes challenging when the task involves completing several sub-tasks sequen-

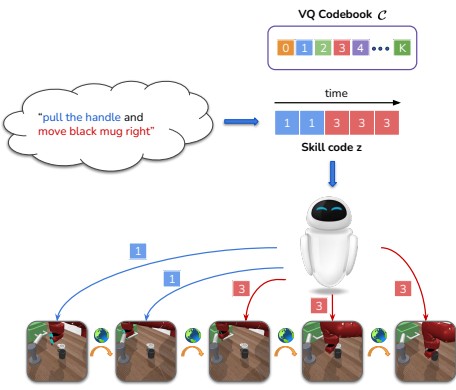

Figure 1: **Overview of LISA.** Given a language instruction, our method learns discrete skill abstractions $z$, picked from a vector codebook $\mathcal{C}$. The policy conditioned on a skill executes distinct behaviors and solves different sub-tasks. See GIF.

---

[*]Equal contribution.

tially, like the example shown in Figure 1. A crucial step towards solving this problem is exploiting the inherent hierarchical structure of natural language. For example, given the task specification "pull the handle and move black mug right", we can split it into learning two independent *primitive behaviors or skills*, *i.e.* "pull the handle" and "move black mug right". If we are able to decompose the problem of solving these complex tasks into learning skills, we can re-use and compose these learned skills to generalize to unseen tasks in the future. This is especially useful in the low-data regime, since *we may not see all possible tasks given the limited dataset, but may see all the constituent sub-tasks*. Using such hierarchical learning, we can utilize language effectively and learn skills as the building blocks of complex behaviors.

Utilizing language effectively to learn skills is a non-trivial problem and raises several challenges. (i) The process of learning skills from language-conditioned trajectories is unsupervised as we may not have knowledge about which parts of the trajectory corresponds to each skill. (ii) We need to ensure that the learned skills are useful, *i.e.* encode behavior that can be composed to solve new tasks. (iii) We would like the learned skills to be interpretable by humans, both in terms of the language and the behaviours they encode. There are several benefits of interpretability. For example, it allows us to understand which skills our model is good at and which skills it struggles with. In safety critical settings such as robotic surgery or autonomous driving, knowing what each skill does allows us to pick and choose which skills we want to run at test time. It also provides a visual window into a neural network policy which is extremely desirable [54]. There have been prior works such as [38, 47, 13] that have failed to address these challenges and condition on language in a monolithic fashion without learning skills. As a result, they tend to perform poorly on long-horizon composition tasks such as the one in Figure 1.

To this end, we propose **L**earning **I**nterpretable **S**kill **A**bstractions from language **(LISA), a hierarchical imitation learning framework that can learn interpretable skills from language-conditioned offline demonstrations.** LISA uses a two-level architecture – a skill predictor that predicts quantized skills from a learnt vector codebook and a policy that uses these skill vector codes to predict actions. The discrete skills learned from language are interpretable (see Figure 2 and 4) and can be composed to solve long-range tasks. Using quantization maximizes skill reuse and enforces a bottleneck to pass information from the language to the policy, enabling unsupervised learning of interpretable skills. We perform experiments on grid world navigation and robotic manipulation tasks and show that our hierarchical method can outperform a strong non-hierarchical baseline based on Decision Transformer [11] in the low-data regime. We analyse these skills qualitatively and quantitatively and find them to be highly correlated to language and behaviour. Finally, using these skills to perform long-range composition tasks on a robotic manipulation environment results in performance that is nearly *2x better* than the non-hierarchical version.

Concretely, our contributions are as follows:

- We introduce LISA, a novel hierarchical imitation framework to solve complex tasks specified via language by learning re-usable skills.

- We demonstrate the effectiveness of our approach in the low-data regime where its crucial to break down complex tasks to generalize well.

- We show our method performs well in long-range composition tasks where we may need to apply multiple skills sequentially.

- We also show that the learned skills are highly correlated to language and behaviour and can easily be interpreted by humans.

## 2 Related Work

### 2.1 Imitation Learning

Imitation learning (IL) has a long history, with early works using behavioral cloning [41–43] to learn policies via supervised learning on expert demonstration data. Recent methods have shown significant improvements via learning reward functions [21] or Q-functions [18] from expert data to mimic expert behavior. Nevertheless, these works typically consider a single task. An important problem here is multi-task IL, where the imitator is trained to mimic behavior on a variety of training tasks with the goal of generalizing the learned behaviors to test tasks. A crucial variable in the multi-task IL set-up is *how the task is specified*, e.g vectorized representations of goal states [37], task IDs [24], and single demonstrations [56, 14, 16, 57]. In contrast, we focus on a multi-task IL setup

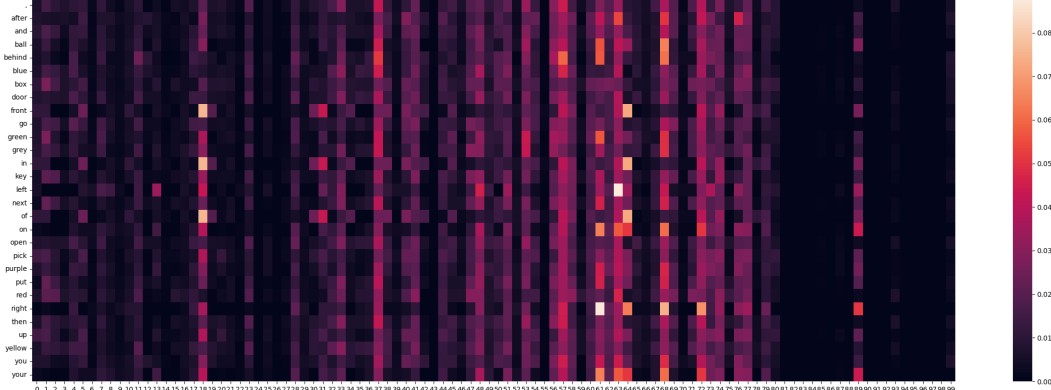

Figure 2: **LISA Skill Heat map.** We show the corresponding word frequencies for 100 different learned skills on BabyAI BossLevel task (column normalized). The x-axis is the skill index and the y-axis is the task vocabulary. LISA's learned skills are interpretable, encode diverse behavior, and are distinctly activated for different words. (zoom in for the best view)

with task-specification through language, one of the most natural and versatile ways for humans to communicate desired goals and intents.

## 2.2 Language Grounding

Several prior works have attempted to ground language with tasks or use language as a source of instructions for learning tasks with varying degrees of success [32, 55, 4, 39, 5]. [27] is a good reference for works combining language with sequential-decision making.

But apart from a few exceptions, most algorithms in this area use the language instruction in a monolithic fashion and are designed to work for simple goals that requires the agent to demonstrate a single skill [40, 9, 20, 6] or tasks where each constituent sub-goal has to be explicitly specified [10, 46, 34, 3, 52, 50, 17, 35, 31]. Some recent works have shown success on using play data [28] or *pseudo-expert* data such as LOReL [38] and CLIPORT [47]. LOReL and CLIPORT are not hierarchical techniques. [28] can be interpreted as a hierarchical technique that generates latent sub-goals as a function of goal images, language instructions and task IDs but the skills learned by LISA are purely a function of language and states alone and do not require goal images or task IDs. [23, 22] and [49] are some examples of works that use a two-level architecture for language conditioned tasks but neither of these methods learn skills that are interpretable.

## 2.3 Latent-models and Hierarchical Learning

Past works have attempted to learn policies conditioned on latent variables and some of them can be interpreted as hierarchical techniques. For example, [15] learns skills using latent variables that visit different parts of the environment's state space. [45] improved on this by learning skills that were more easily predictable using a dynamics model. But these fall more under the category of skill discovery than hierarchical techniques since the skill code is fixed for the entire trajectory, as is the case with [15]. [29] and [26] are other works that use a latent-variable approach to IL. But these approaches don't necessarily learn a latent variable with the intention of breaking down complex tasks into skills. With LISA, we sample several skills per trajectory with the clear intention of each skill corresponding to completing a sub-task for the whole trajectory. Also, none of the methods mentioned here condition on language.

There has been some work on hierarchical frameworks for RL to learn high-level action abstractions, called *options* [51], such as [25, 58, 36] but these works are not goal-conditioned. Unlike LISA, these works don't use language and the options might lack diversity and not correspond to any concrete or interpretable skills. Furthermore, none have used the VQ technique to learn options and often suffer from training instabilities.

# 3 Approach

The key idea of LISA is to learn quantized skill representations that are informative of both language and behaviors, which allows us to break down high-level instructions, specified via language, into discrete, interpretable and composable codes (see Fig. 2, Fig. 6 and Fig. 8 for visualizations). These codes enable learning explainable and controllable behaviour, as shown in Fig. 1 and Fig. 4.

Section 3.1 describes the problem formulation, an overview of our framework, and presents our language-conditioned model. Section 3.2 provides details on the training approach.

## 3.1 Language-conditioned Skill Learning

### 3.1.1 Problem Setup

We consider general multi-task environments, represented as a task-augmented Markov decision process (MDP) with a family of different tasks $\mathcal{T}$. A task $\mathcal{T}_i$ may be composed of other tasks in $\mathcal{T}$ and encode multiple sub-goals. For example, in a navigation environment, a task could be composed of two or more sub-tasks - "pick up ball", "open door" - in any hierarchical order. $\mathcal{S}, \mathcal{A}$ represent state and action spaces. We assume that each full task has a *single* natural language description $l \in L$, where $L$ represents the space of language instructions. Any sub-goals for the task are encoded within this single language instruction.

We assume access to an offline dataset $\mathcal{D}$ of trajectories obtained from an optimal policy for a variety of tasks in an environment with only their language description available. Each trajectory $\tau^i = (l^i, \{(s_1^i, a_1^i), (s_2^i, a_2^i), ..., (s_T^i, a_T^i)\})$ consists of the language description and the observations $s_t^i \in \mathcal{S}$, actions $a_t^i \in \mathcal{A}$ taken over $T$ timesteps. The trajectories are not labeled with any rewards.

Our aim is to predict the expert actions $a_t$, given a language instruction and past observations.

Note that each trajectory in the training dataset can comprise of any number of sub-tasks. For example, we could have a trajectory to "open a door" and another to "pick up a ball and close the door" in the training data. With LISA we aim to solve the task "open a door and pick up the ball" at test time even though we haven't seen this task at training time. In a trajectory with multiple sub-tasks, the training dataset **does not** give us information about where one sub-task ends and where another one begins.

LISA must learn how to identify and stitch together these sub-tasks learned during training, in order to solve a new language instruction such as the one shown in Fig. 1 at test time.

### 3.1.2 Hierarchical Skill Abstractions

We visualize the working of LISA in Figure 3. Our framework consists of two modules: a skill predictor $f : L \times \mathcal{S} \rightarrow \mathcal{C}$ and a policy $\pi : \mathcal{S} \times \mathcal{C} \rightarrow \mathcal{A}$. Here, $\mathcal{C} = \{z^1, \ldots, z^K\}$ is a learnable codebook of $K$ quantized skill latent codes. $D$ is the dimension of the latent space of skills.

Our key idea is to break learning behavior from language in two stages: 1) Learn discrete latent codes $z$, representing skills, from the full-language instruction to decompose the task into smaller sub-goals 2) Learn a policy $\pi$ conditioned only on these discrete codes. In LISA, both stages are trained end-to-end.

Given an input $\tau = (l, \{s_t, a_t\}_{t=1}^T)$, the skill predictor $f$ predicts a skill code $\tilde{z} \in \mathcal{R}^D$ at a timestep $t$ as $\tilde{z} = f(l, (s_t, s_{t-1}, ...))$. These codes are discretized using a vector quantization operation $\mathbf{q}(\cdot)$ that maps a latent $\tilde{z}$ to its closest codebook entry $z = \mathbf{q}(\tilde{z})$. The quantization operation $\mathbf{q}(\cdot)$ helps in learning discrete skill codes and acts as a bottleneck on passing language information. We detail its operation in Sec. 3.2.

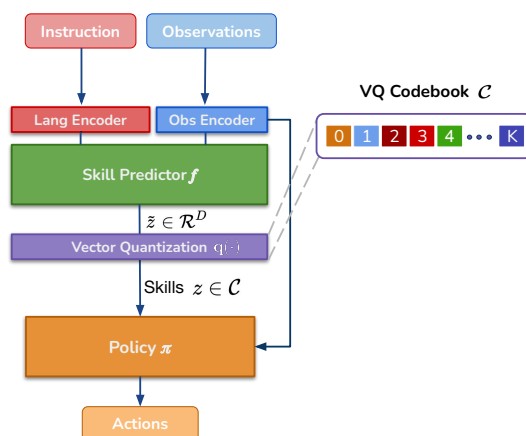

Figure 3: **LISA Architecture**: The skill predictor $f$ gets the language instruction and a sequence of observations as the input, processed through individual encoders. It predicts quantized skill codes $z$ using a learnable cookbook $\mathcal{C}$, that encodes different sub-goals, and passes them to the policy $\pi$. LISA is trained end-to-end.

The chosen skill code $z$, is persisted for $H$ timesteps where $H$ is called the horizon. More details on how we chose the horizon and ablations studies on the choice of $H$ can be found in appendix sections D and F.1. After $H$ timesteps, the skill predictor is invoked again to predict a new skill. This enforces the skill to act as a temporal abstraction on actions, i.e. options [51]. The policy $\pi$ predicts the action $a_t$ at each timestep $t$ conditioned on the state and a single skill code $z$ that is active at that timestep. For $\pi$ to correctly predict the original actions, it needs to use the language information encoded in the skill codes.

LISA learns quantized skill codes in a vector codebook instead of continuous embeddings as this encourages reusing and composing these codes together to pass information from the language input to the actual behavior. Our learnt discrete skill codes adds interpretability and controllability to the policy's behavior.

## 3.2 Training LISA

**Learning Discrete Skills.** LISA uses Vector Quantization (VQ), inspired from [53]. It is a natural and widely-used method to map an input signal to a low-dimensional discrete learnt representation. VQ learns a codebook $\mathcal{C} \in \left\{ z^1, \ldots, z^K \right\}$ of $K$ embedding vectors. Given an embedding $\tilde{z}$ from the skill predictor $f$, it maps the embedding to the closest vector in the codebook:

$$z = \mathbf{q}(\tilde{z}) =: \underset{z^k \in \mathcal{C}}{\arg\min} \|\tilde{z} - z^k\|_2$$

with the codebook vectors updated to be the moving average of the embeddings $z$ closest to them. This can be classically seen as learning $K$ cluster centers via $k$-means [19].

Backpropagation through the non-differentiable quantization operation is achieved by a straight-through gradient estimator, which simply copies the gradients from the decoder to the encoder, such that the model and codebook can be trained end-to-end.

VQ enforces each learnt skill $z$ to lie in $\mathcal{C}$, which can be thought as learning $K$ prototypes or cluster centers for the language embeddings using the seen states. This acts as a bottleneck that efficiently decomposes a language instruction into sub-parts encoded as discrete skills.

**LISA Objective.** LISA is trained end-to-end using an objective $\mathcal{L}_{\text{LISA}} = \mathcal{L}_{\text{BC}} + \lambda \mathcal{L}_{\text{VQ}}$, where $\mathcal{L}_{\text{BC}}$ is the behavior-cloning loss on the policy $\pi_\theta$, $\lambda$ is the VQ loss weight and $\mathcal{L}_{\text{VQ}}$ is the vector quantization loss on the skill predictor $f_\phi$ given as:

$$\mathcal{L}_{\text{VQ}}(f) = \mathbb{E}_\tau[\|\text{sg}\left[\mathbf{q}(\tilde{z})\right] - \tilde{z}\|_2^2] \tag{1}$$

with $\tilde{z} = f_\phi(l, (s_t, s_{t-1}, ..))$.

sg $[\cdot]$ denotes the stop-gradient operation. $\mathcal{L}_{\text{VQ}}$ is also called *commitment loss*. It minimizes the conditional entropy of the skill predictor embeddings given the codebook vectors, making the embeddings stick to a single codebook vector.

The codebook vectors are learnt using an exponential moving average update, same as [53].

**Avoiding language reconstruction.** LISA avoids auxiliary losses for language reconstruction from the skills latent codes and it's not obvious why the skill codes are properly encoding language, and we expand on it here.

For a given a signal $X$ and a code $Z$, reconstructing the signal from the code as $\tilde{X} = f(Z)$ using cross-entropy loss amounts to maximizing the Mutual Information (MI) $I(X, Z)$ between $X$ and $Z$ [1, 7]. In our case, we can write the MI between the skill codes and language using entropies as: $I(z, l) = H(z) - H(z \mid l)$, whereas methods that attempt to reconstruct language apply the following decomposition: $I(z, l) = H(l) - H(l \mid z)$. Here, $H(l)$, the entropy of language instructions, is constant, and this gives us the cross-entropy loss.

Thus we can avoid language reconstruction via cross-entropy loss by maximizing $I(z, l)$ directly. In LISA, $\mathcal{L}_{\text{vq}} = -H(z \mid l)$, and we find there is no need to place a constraint on $H(z)$ as the learned skill codes are diverse, needing to encode enough information to correctly predict the correct actions.[1]

---

[1]In experiments, we tried enforcing a constraint on $H(z)$ by using extra InfoNCE loss term but don't observe any gains.

**Algorithm 1** Training LISA

**Input:** Dataset $\mathcal{D}$ of language-paired trajectories
**Input:** Num skills $K$ and horizon $H$
1: Initialize skill predictor $f_\phi$, policy $\pi_\theta$
2: Vector Quantization op $\mathbf{q}(\cdot)$
3: **while** *not converged* **do**
4:     Sample $\tau = (l, \{s_0, s_1, s_2...s_T\}, \{a_0, a_1, a_2...a_T\})$
5:     Initialize $S = \{s_0\}$                                        ▷ List of seen states
6:     **for** $k = 0..\lfloor \frac{T}{H} \rfloor$ **do**                      ▷ Sample a skill every H steps
7:         $z \leftarrow \mathbf{q}(f_\phi(l, S))$
8:         **for** *step* $t = 1..H$ **do**     ▷ Predict actions using a fixed skill and context length $H$
9:             $a_{kH+t} \leftarrow \pi_\theta(z, S[: -H])$
10:             $S \leftarrow S \cup \{s_{kH+t}\}$                         ▷ Append seen state
11:         **end for**
12:         Train $f_\phi, \pi_\theta$ using objective $\mathcal{L}_{\text{LISA}}$
13:     **end for**
14: **end while**

$z = 14$

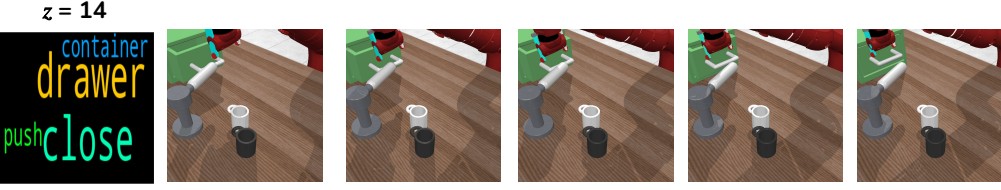

Figure 4: **Behavior with fixed LISA options.** We show the word clouds and the behavior of the policy obtained by using a fixed skill code $z = 14$ for an entire episode. We find that this code encodes the skill "closing the drawer", as indicated by the word cloud. The policy executes this skill with a high degree of success when conditioned on this code for the entire trajectory, across multiple environment initializations and seeds.

As a result, LISA can maximize the MI between the learnt skills and languages without auxiliary reconstruction losses and enforcing only $\mathcal{L}_{\text{vq}}$ on the skill codes. We empirically estimate the MI between the language and skill codes and find that our experiments confirm this in Section 4.6.

### 3.2.1 LISA Implementation

LISA can be be implemented using different network architectures, such as Transformers or MLPs.

In our experiments, we use Transformer architectures with LISA, but we find that out method is effective even with simple architectures choices such as MLPs, as shown in the appendix section F.5. Even when using Transformers for both the skill predictor and the policy network, our compute requirement is comparable to the non-hierarchical Flat Transformer policy as we can get away with using fewer layers in each module.

**Language Encoder.** We use a pre-trained DistilBERT [44] encoder to generate language embeddings from the text instruction. We fine-tune the language encoder end-to-end and use the full language embedding for each word token, and not a pooled representation of the whole text.

**Observation Encoder.** For image observations, we use convolution layers to generate embeddings. For simple state representations, we use MLPs.

**Skill Predictor.** The skill predictor network $f$ is implemented as a small Causal Transformer network that takes in the language embeddings and the observation embeddings at each time step. The language embeddings are concatenated at the beginning of the observation embeddings before being fed into the skill predictor. The network applies a causal mask hiding the future observations.

**Policy Network.** Our policy network $\pi$, also implemented as a small Causal Transformer inspired by Decison Transformer (DT) [11]. However, unlike DT, our policy is not conditioned on any reward signal, but on the skill code. The sequence length of $\pi$ is the horizon $H$ of the skills which is much smaller compared to the length of the full trajectory.

**Flat Decision Transformer Baseline.** Our flat baseline is based on DT and is implementation-wise similar to LISA, but without a skill predictor network. The policy here is a Causal Transformer, where

we modify DT to condition on the language instruction embedding from a pre-trained DistillBERT text encoder instead of the future sum of returns. We found this baseline to be inefficient at handling long-range language instructions, needing sequence lengths of 1000 on complex environments such as BabyAI-BossLevel in our experiments.

Since LISA has two transformers as opposed to just one in the flat baseline we ensured that the baseline and our method had a similar number of total parameters. To this end, the flat baseline uses Transformer network with 2 self-attention layers, and LISA's skill predictor and policy use Transformer network with a single self-attention layer each. We also ensured that the embedding dimension and the number of heads in each layer were exactly the same in both LISA and the flat baseline. Details of this are provided in appendix sections D.1 and D.2 respectively. In fact, one could argue that LISA has less representation power because the policy transformer can only attend to the last H steps while the flat baseline can attend to the entire trajectory which is what makes it an extremely strong baseline. The flat baseline also uses the same pre-trained DistillBERT text encoder model as LISA for dealing with natural language input.

## 4  Experiments

In this section, we evaluate LISA on grid-world navigation and robotic manipulation tasks. We compare the performance of LISA with a strong non-hierarchical baseline in the low-data regime. We then analyse our learnt skill abstractions in detail – what they represent, how we can interpret them and how they improve performance on downstream composition tasks.

For the sake of brevity, we present additional ablations in the Appendix F, on doing manual planning with LISA skills (Section F.2), transferring learned skills to different environments (Section F.3) and learning continuous skills (Section F.6).

### 4.1  Datasets

Several language-conditioned datasets have been curated as of late such as [46, 48, 13, 38, 33, 2, 10, 12]. Nevertheless, a lot of these datasets focus on complex-state representations and navigation in 3D environments, making them challenging to train on and qualitatively analyze our skills as shown in Fig. 4. We found BabyAI, a grid-world navigation environment and LOReL, a robotic manipulation environment as two diverse test beds that were very different from each other and conducive for hierarchical skill learning as well as detailed qualitative and quantitative analysis of our learned skills and we use them for our experiments.

**BabyAI Dataset.** The BabyAI dataset [13] contains 19 levels of increasing difficulty where each level is set in a grid world and an agent sees a partially observed ego-centric view in a square of size 7x7. The agent must learn to perform various tasks of arbitrary difficulty such as moving objects between rooms, opening or closing doors, etc. all with a partially observed state and a language instruction. The language instructions for easy levels are quite simple but get exponentially more challenging for harder levels and contain several skills that the agent must complete in sequence (examples in appendix section C.1). The dataset provides 1 million expert trajectories for each of the 19 levels, but we use $0.1 - 10\%$ of these trajectories to train our models. We evaluate our policy on a set of 100 different instructions from the gym environment for each level, which contain high percentage of unseen environments layouts and language instructions given the limited data we use for training. More details about this dataset can be found in Appendix C.1 and in the BabyAI paper.

**LOReL Sawyer Dataset.** This dataset [38] consists of *pseudo-expert* trajectories or *play data* collected from a replay buffer of a random RL policy and has been labeled with post-hoc crowd-sourced language instructions. Hence, the trajectories complete the language instruction provided but may not necessarily be optimal. Play data is inexpensive to collect [30] in the real world and it is important for algorithms to be robust to such datasets as well. However, due to the randomness in the trajectories, this makes the dataset extremely difficult to use in a behavior cloning (BC) setting. Despite this, we are able to achieve good performance on this benchmark and are able to learn some very useful skills. The LOReL Sawyer dataset contains 50k trajectories of length 20 on a simulated environment with a Sawyer robot. We evaluate on the same set of 6 tasks that the original paper does for our results in Table 1: *close drawer, open drawer, turn faucet right, turn faucet left, move black mug right, move white mug down*. We use two different settings - with robot state space observations and partially-observed image observations. More details can be found in the Appendixd C.2 and in the LOReL paper.

Table 1: **Imitation Results:** We show our success rates (in %) compared to the original method and a flat non-hierarchical Decision Transformer baseline on each dataset over 3 seeds. LISA outperforms all other methods in the low-data regime, and reaches similar performance as the number of demonstrations increases. Best method shown in **bold**.

| Task | Num Demos | Original | Lang DT | LISA |
|------|-----------|----------|---------|------|
| BabyAI GoToSeq | $1k$ | $33.3 \pm 1.3$ | $49.3 \pm 0.7$ | $\mathbf{59.4 \pm 0.9}$ |
| BabyAI GoToSeq | $10k$ | $40.4 \pm 1.2$ | $62.1 \pm 1.2$ | $\mathbf{65.4 \pm 1.6}$ |
| BabyAI GoToSeq | $100k$ | $47.1 \pm 1.1$ | $74.1 \pm 2.3$ | $\mathbf{77.2 \pm 1.7}$ |
| BabyAI SynthSeq | $1k$ | $12.9 \pm 1.2$ | $42.3 \pm 1.3$ | $\mathbf{46.3 \pm 1.2}$ |
| BabyAI SynthSeq | $10k$ | $32.6 \pm 2.5$ | $52.1 \pm 0.5$ | $\mathbf{53.3 \pm 0.7}$ |
| BabyAI SynthSeq | $100k$ | $40.4 \pm 3.3$ | $\mathbf{64.2 \pm 1.3}$ | $61.2 \pm 0.6$ |
| BabyAI BossLevel | $1k$ | $20.7 \pm 4.6$ | $44.5 \pm 3.3$ | $\mathbf{49.1 \pm 2.4}$ |
| BabyAI BossLevel | $10k$ | $28.9 \pm 1.3$ | $\mathbf{60.1 \pm 5.5}$ | $58 \pm 4.1$ |
| BabyAI BossLevel | $100k$ | $45.3 \pm 0.9$ | $\mathbf{72.0 \pm 4.2}$ | $69.8 \pm 3.1$ |
| LOReL - States (fully obs.) | $50k$ | $6 \pm 1.2^*$ | $33.3 \pm 5.6$ | $\mathbf{66.7 \pm 5.2}$ |
| LOReL - Images (partial obs.) | $50k$ | $29.5 \pm 0.07$ | $15 \pm 3.4$ | $\mathbf{40 \pm 2.0}$ |

$^*$ We optimized a language-conditioned BC model following the LOReL paper to the best of our abilities but could not get better performance.

## 4.2 Baselines

**Original.** These refer to the baselines from the original paper for each dataset. For BabyAI, we trained their non-hierarchical RNN based method on different number of trajectories. Similarly, on LOReL we compare with the performance of language-conditioned BC. The original LOReL method uses a planning algorithm on a learned reward function to get around the sub-optimal nature of the trajectories. We found the BC baseline as a more fair comparison, as LISA is trained using BC as well. Nonetheless, we compare with the original LOReL planner in Section 4.7 for composition tasks. LOReL results in Table 1 refer to the performance on the 6 *seen* instructions in the LOReL evaluation dataset, same as ones reported in the original paper.

**Flat Baseline.** We implement a non-hierarchical baseline using language-conditioned Decision Transformer denoted as **Lang DT**, the details of which are in section 3.2.1.

## 4.3 How does performance of LISA compare with non-hierarchical baselines in low-data regime?

We consider three levels from the BabyAI environment and the LOReL Sawyer environment. For BabyAI, we consider the GoToSeq, SynthSeq and BossLevel tasks since they are challenging and require performing several sub-tasks one after the other. Since these levels contain instructions that are compositional in nature, when we train on limited data the algorithm must learn skills which form complex instructions to generalize well to unseen instructions at test time.

Our experimental results are shown in Table 1. We train the models on a randomly sampled 1k, 10k and 100k trajectories from the full BabyAI dataset and 50k trajectories on the LOReL dataset. We use more data from the LOReL dataset because of the sub-optimal nature of the trajectories. **On all the environments, our method is competitive to or outperforms the strong non-hierarchical Decision Transformer baseline.**

The gap grows larger as we reduce the number of trajectories trained on, indicating that our method is able to leverage the common sub-task structures better and glean more information from limited data. As expected, with larger amounts of training data it becomes hard to beat the flat baseline since the model sees more compositions during training and can generalize better at test time [8]. As mentioned above, we evaluate on the same 6 *seen* instructions the original LOReL paper did. We also evaluate the performance on varying language instructions on LOReL, similar to the original paper, with additional results in Appendix E.

We were pleasantly surprised that **LISA is 2x better than the flat Lang-DT baseline on LOReL tasks**, reaching $40\%$ success rate using partial image observations despite the sub-optimal nature of the data. One explanation for this is that the discrete skill codes are able to capture *different ways of doing the same task*, thereby allowing LISA to *learn an implicit multi-modal policy*. This is not possible with the flat version as it has no way to compartmentalize these noisy trajectories, and perhaps tends to overfit on this noisy data, leading to performance degradation.

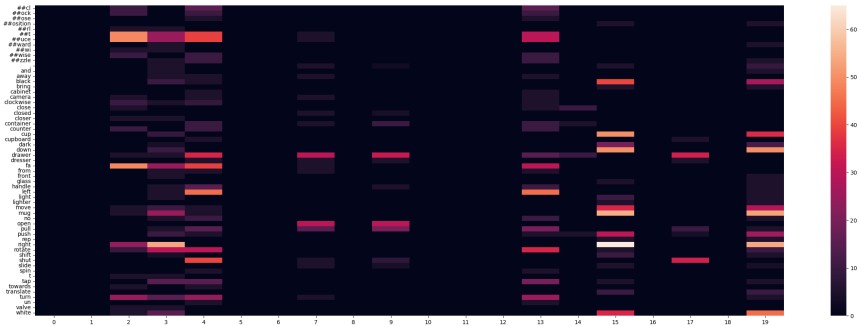

Figure 5: **LISA Skill Heat map on LOReL.** We show the corresponding word frequency for 20 learned skills on LOReL (column normalized). The sparsity and the bright spots show that specific skills correspond to specific language tokens. (zoom in for best view)

## 4.4 What skills does LISA learn? Are they diverse?

To answer this question, we analyse the skills produced by LISA and the language tokens corresponding to each skill. We plot a heat map in Figure 5 corresponding to the correlation between the language tokens and skill codes. Here, we plot the map corresponding to the LOReL dataset. From the figure, we can see that certain skill codes correspond very strongly to certain language tokens and by extension, tasks. We also see the sparse nature of the heat maps which indicates that each skill corresponds to distinct language tokens. We also plot word clouds corresponding to four different options in the LOReL environment in Figure 6 and we notice that different options are triggered by different language tokens. From the figure, it is clear that the skill on the top left corner corresponds to *close the drawer* and the skill on the top right corresponds to *turn faucet left*. Similar word clouds and heat maps for the BabyAI environments are in the appendix section B.3.

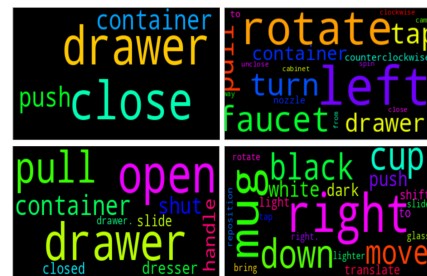

Figure 6: **Word clouds on LOReL**: We show the most correlated words for 4 different learnt skill codes on LOReL. We can see that the codes represent interpretable and distinguishable skills. For e.g, the code on the top left corresponds to closing the drawer. (note that container is a synonym for drawer in the LOReL dataset)

## 4.5 Do the skills learned by LISA correspond to interpretable behavior?

We have seen that the different skills correspond to different language tokens, but do the policies conditioned on these skills behave according to the language tokens? To understand this, we fix the skill code for the entire trajectory and run the policy i.e. we are shutting off the skill predictor and always predicting the same skill for the entire trajectory. As we can see from the word cloud and the corresponding trajectory in Figure 4, the behaviour for skill code 14 is exactly what we can infer from the language tokens in the word cloud – *close the drawer*. More such images and trajectories can be found in the appendix section B.5.

## 4.6 Why do LISA learned skills show such a strong correlation to language?

As mentioned in section 3.2, the *commitment loss* from VQ acts as a way to increase the MI between the language and the skill codes during training. This allows the codes to be highly correlated with language without any reconstruction losses. To analyze this, we plot the MI between the options and the language during training on the BabyAI BossLevel with 1k trajectories and the plot can be seen in Figure 7. The plots show the MI increasing over training for a wide range of

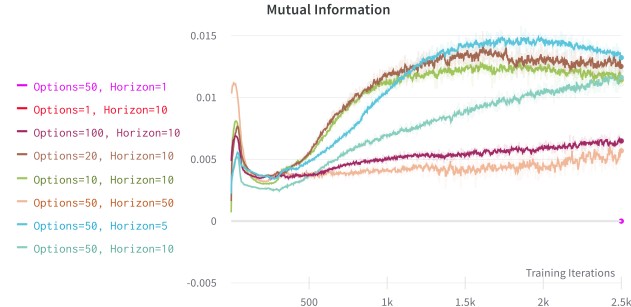

Figure 7: **MI between language and skill codes:** We show the Mutual Information over training iterations for various settings of LISA on the BabyAI BossLevel environment

settings as we vary the number of skills and the horizon. In the ablation studies below, we report the success rate corresponding to each of these curves and we notice that there's almost a direct correlation with increasing MI and task performance. This is very encouraging since it clearly shows that the skills are encoding language and that directly impacts the performance of the behavior cloning policy.

### 4.7 Can we use the learned skills to perform new composition tasks?

To test our composition performance, we evaluate on LOReL composition tasks using images in Table 2. To this end, we handcraft 15 **unseen** composition instructions. We have listed these instructions in the Appendix Table 5 with one such example *"pull the handle and move black mug down"*. We

Table 2: **LISA Composition Results:** We show our performance on the LOReL Sawyer environment on 15 unseen instructions compared to baselines

| Method | Success Rate (in %) |
|---|---|
| Flat | $13.33 \pm 1.25$ |
| LOReL Planner | $18.18 \pm 1.8$ |
| LISA (Ours) | $\mathbf{20.89 \pm 0.63}$ |

ran 10 different runs of each instruction across 3 different seeds. As we can see, our performance is nearly 2x that of the non-hierarchical baseline. We also compare with the original LOReL planner on these composition tasks and we notice that we perform slightly better despite them having access to a reward function and a dynamics model pre-trained on 1M frames while LISA is trained from scratch. We set the max number of episode steps to 40 from the usual 20 for all the methods while performing these experiments because of the compositional nature of the tasks.

Note that results in Table 1 show compositionality performance on the BabyAI dataset as we train with 0.1%-10% of the data. When we evaluate on the gym environment generating any possible language instruction from the BabyAI grammar, we may

Table 3: **BabyAI:** % of unseen instructions at test time for different training regimes.

| Environment | 1k trajs | 10k trajs | 100k trajs |
|---|---|---|---|
| GoToSeq | 76% | 63.8% | 48% |
| SynthSeq | 76.3% | 66% | 50.7% |
| BossLevel | 77.3% | 66.9% | 51.3% |

come across several unseen instructions at test time. To give a sense of the % of unseen instructions for BabyAI when we evaluate on the gym environment, we take the different BabyAI environments and report the % of unseen language instructions seen at test time for different training data regimes in Table 3. For each statistic, we sample 10,000 random instructions from the environment and check how many are unseen in the training dataset used, repeated over 3 different seeds.

### 4.8 How does LISA compare to simply doing K-Means clustering on the language and state embeddings?

In LISA, the VQ approach can be seen as taking the concatenated language-state inputs and projecting it into a learned embedding space. VQ here simply learns $K$ embedding vectors that act as $K$ cluster centers for the projected input vectors in this embedding space, and allows for differentiability, enabling learning through backpropagation. This is also similar to propotypical methods used for few-shot learning and allows for deep differentiation clustering, giving an intuition of why it works.

To compare against $k$-means, we construct a simple unsupervised learning baseline that clusters trajectories in the training dataset using $k$-means. Specifically, in the BabyAI BossLevel environment using 1k training trajectories, we take all concatenated language-state vectors for all trajectories in the dataset and cluster them using $k$-means and use the assigned cluster centers as the skill codes. We then learn a policy using these skill codes to measure their efficacy and found that LISA has a performance of $\mathbf{49.1 \pm 2.4}\%$ and $k$-means has a performance of $20.2 \pm 5.2\%$ over 3 seeds.

Thus, we see that using the simple $k$-means skills is insufficient to learn a good policy to solve the BabyAI BossLevel task, as the skills are not representative enough of the language instructions. A reason for this is that language and state vectors lie in different embedding spaces, and K-means based on euclidean distance is not optimal on the concatenated vectors.

## 5 Limitations and Future Work

We present LISA, a hierarchical imitation learning framework that can be used to learn interpretable skill abstractions from language-conditioned expert demonstrations. We showed that the skills are diverse and can be used to solve long-range language tasks and that our method outperforms a strong non-hierarchical baseline in the low-data regime.

However, there are several limitations to LISA and plenty of scope for future work. One limitation of LISA is that there are several hyperparameters to tune that may affect performance like the number of options and the horizon for each option. It certainly helps to have a good idea of the task to decide these hyperparameters even though the ablations show that the method is fairly robust to these choices. Its also useful to learn the horizon for each skill by learning a termination condition and we leave this for future work.

Although our method has been evaluated on the language-conditioned imitation learning setting, its not difficult to modify this method to make it work for image goals or demos, and in the RL setting as well. Its interesting to see if the vector quantization trick can be used to learn goal-conditioned skills in a more general framework.

## Acknowledgments and Disclosure of Funding

We are thankful to John Schulman, Chelsea Finn, Karol Hausman and Dilip Arumugam for initial discussions regarding our method, and to Suraj Nair for providing help with the LOReL baseline.

This research was supported in part by NSF (#1651565, #1522054, #1733686), ONR (N00014-19-1-2145), AFOSR (FA9550-19-1-0024), FLI and Samsung.

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
