# Appendix

# A  Broader Societal Impact

We introduce a new method for language-conditioned imitation learning to perform complex navigation and manipulation tasks. Our intention is for this algorithm to be used in a real-world setting where humans can provide natural language instructions to robots that can carry them out. However, we must ensure that the language commands that we provide to these agents must be well aligned with the objectives of humans and must ensure that we are aware of what actions the agent could take given said command.

# B  More visualizations

## B.1  Generating heat maps and word clouds

To generate heat maps and word clouds, for each evaluation instruction, we run the model and record all the skill codes used in the trajectory generated. We now tokenize the instruction and for each skill code used in the trajectory, record *all* the tokens from the language instruction. Once we have this mapping from skills to tokens, we can plot heat maps and word clouds. This is the best we can do since we don't know exactly which tokens in the instruction correspond to the skills chosen. Therefore, these plots can tend to be a little noisy but we still see some clear patterns. Especially in BabyAI, since the vocabulary is small, we see that several skills correspond to the same tokens because many instructions contain the same tokens. But in LOReL because each task uses almost completely different words, we can see a very sparse heat map with clear correlations.

For reader viewability and aiding the interpretability on the LISA skills, we show below the unnormalized heatmaps showing the skill-word correlations, the column normalized heatmaps showing word frequencies for each skill as well as the row normalized showing the skill frequencies for each word.

## B.2  BabyAI

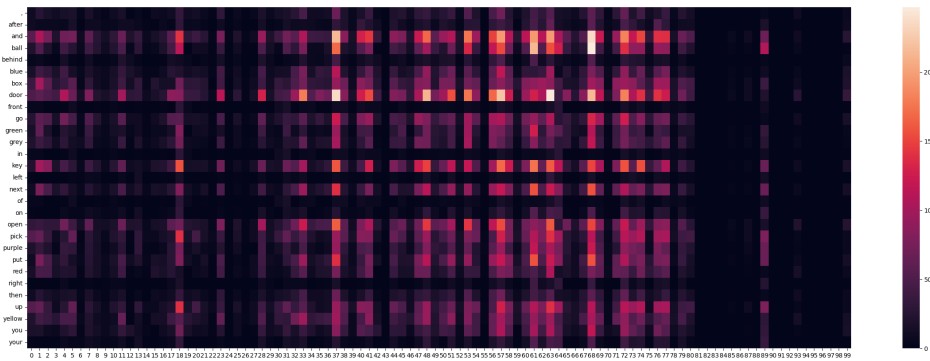

Figure 8: Skill Heat map on BabyAI BossLevel

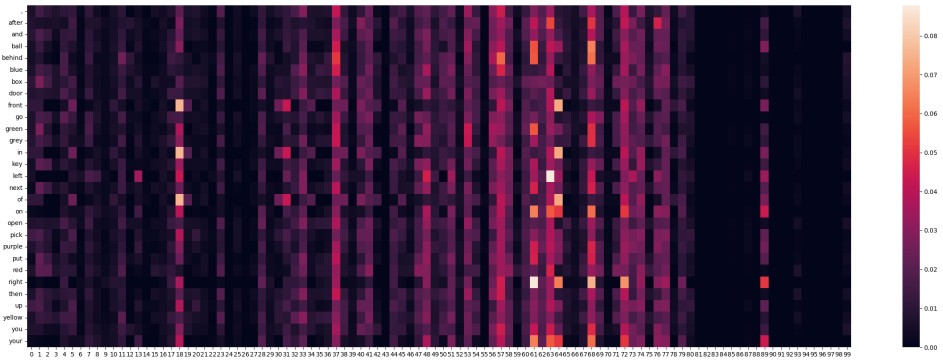

Figure 9: Word Freq. for each skill on BabyAI BossLevel (column normalized)

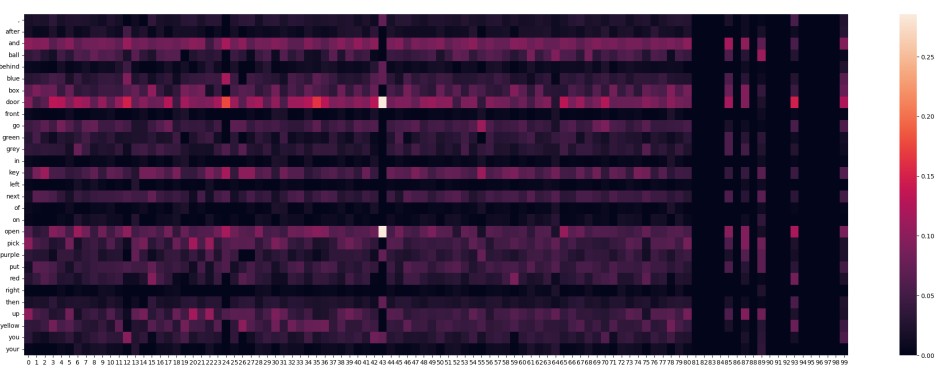

Figure 10: Skill Freq. for each word on BabyAI BossLevel (row normalized)

## B.3 WordClouds

Due to the small vocabulary in BabyAI environment, its hard to generate clean word clouds, nevertheless, we hope they help with interpreting LISA skills.

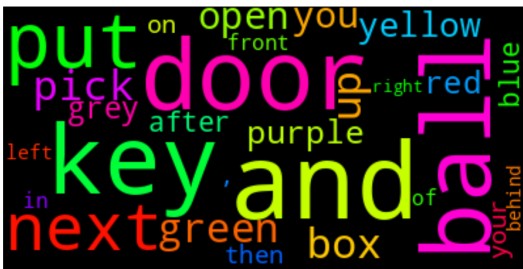

Figure 11: Word Cloud on BabyAI BossLevel for $z = 1$

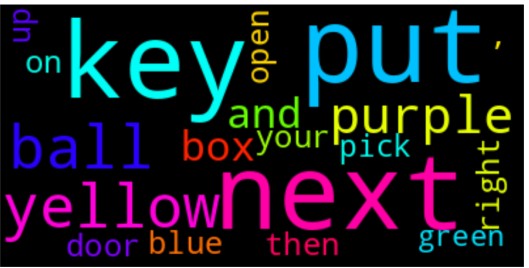

Figure 12: Word Cloud on BabyAI BossLevel for $z = 13$

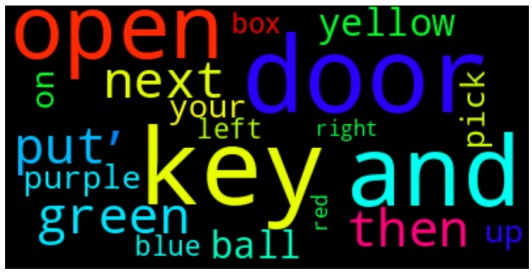

Figure 13: Word Cloud on BabyAI BossLevel for $z = 37$

## B.4  LOReL Sawyer

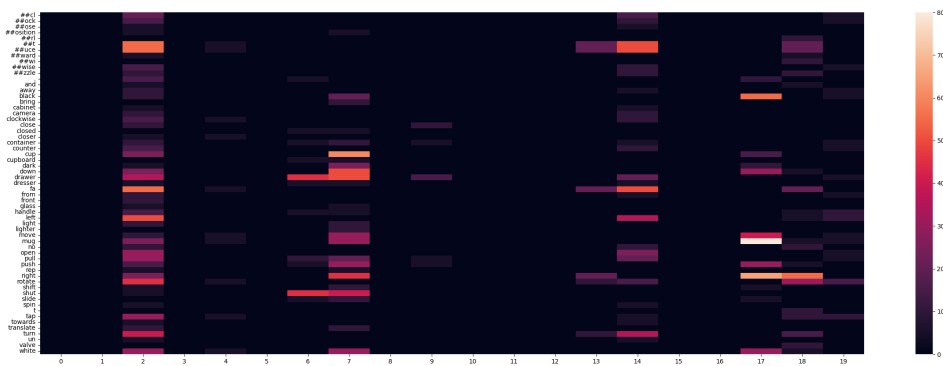

Figure 14: Skill Heat map on LOReL Sawyer

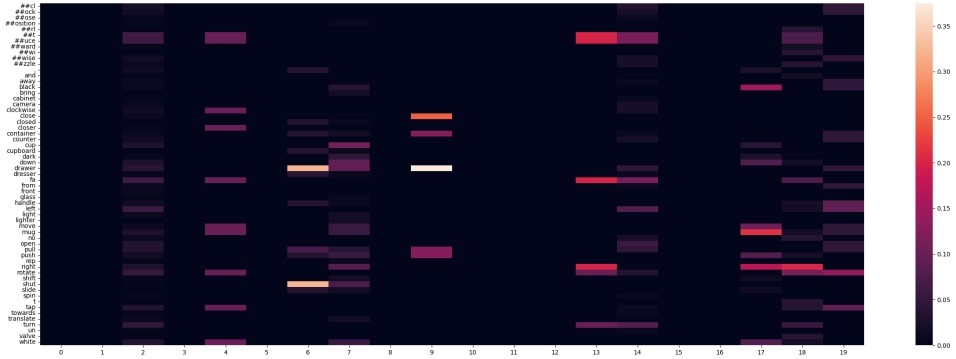

Figure 15: Word Freq. for each skill on LOReL Sawyer (column normalized)

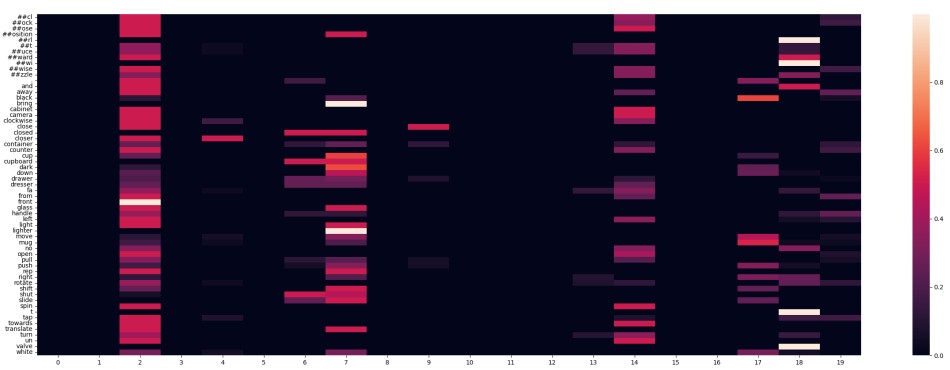

Figure 16: Skill Freq. for each word on LOReL Sawyer (row normalized)

## B.5 Behavior with fixed skills

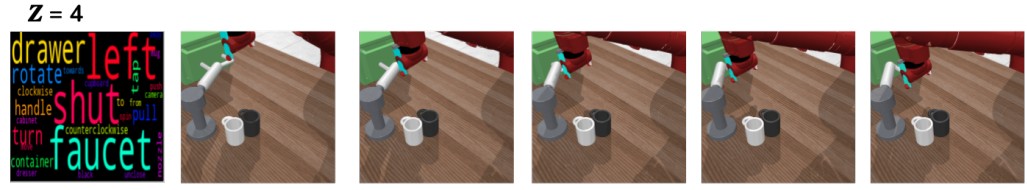

Figure 17: Behavior and language corresponding to skill code 4: "turn faucet left"

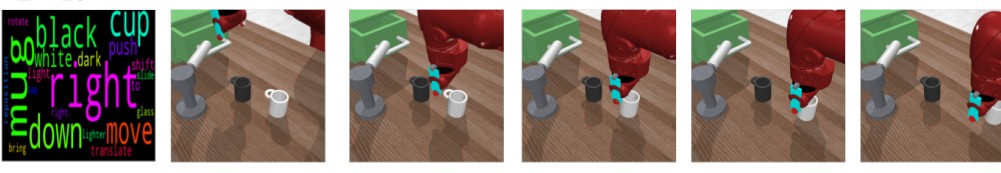

Figure 18: Behavior and language corresponding to skill code 15: "move white mug right"

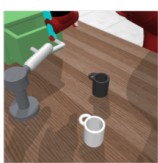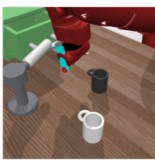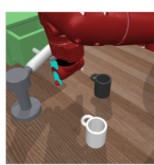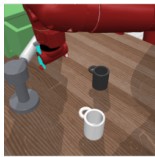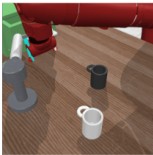

Figure 19: LOReL Composition task: "close drawer and turn faucet left"

## C    Datasets

### C.1    BabyAI Dataset

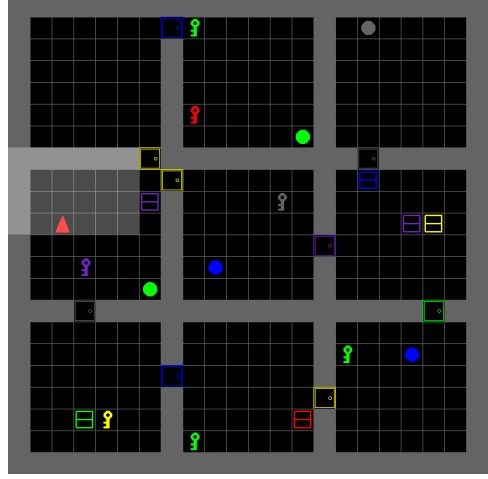

Figure 20: BabyAI BossLevel

The BabyAI dataset [13] contains 19 levels of increasing difficulty where each level is set in a grid world where an agent has a partially observed state of a square of side 7 around it. The agent must learn to perform various tasks of arbitrary difficulty such as moving objects between rooms, opening doors or closing them etc. all with a partially observed state and a language instruction.

Each level comes with 1 million language conditioned trajectories, and we use a small subset of these for our training. We evaluate our model on the environment provided with each level that generates a new language instruction and grid randomly.

We have provided details about the levels we evaluated on below. More details can be found in the original paper.

### C.1.1    GoToSeq

Sequencing of go-to-object commands.
Example command: *"go to a box and go to the purple door, then go to the grey door"*
Demo length: $72.7 \pm 52.2$

### C.1.2 SynthSeq

Example command: *"put a purple key next to the yellow key and put a purple ball next to the red box on your left after you put a blue key behind you next to a grey door"*
Demo length: $81.8 \pm 61.3$

### C.1.3 BossLevel

Example command: *"pick up a key and pick up a purple key, then open a door and pick up the yellow ball"*
Demo length: $84.3 \pm 64.5$

### C.2 LOReL Sawyer Dataset

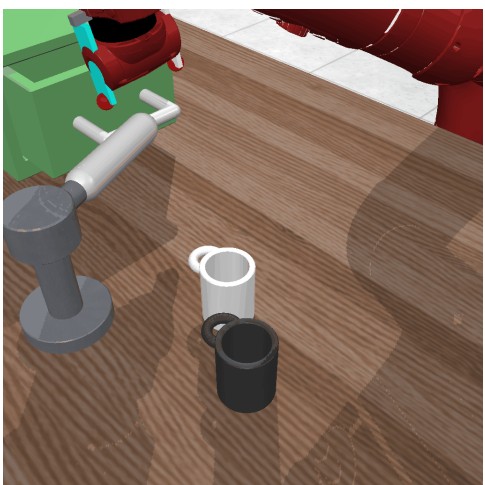

Figure 21: LOReL Sawyer Environment

This dataset [38] consists of *pseudo-expert* trajectories collected from a RL buffer of a a random policy and has been labeled with post-hoc crowd-sourced language instructions. Therefore, the trajectories complete the language instruction provided but may not necessarily be optimal. The Sawyer dataset contains 50k language conditioned trajectories on a simulated environment with a Sawyer robot of demo length 20.

We evaluate on the same set of instructions the original paper does for 1, which can be found in the appendix of the original paper. These consist of the following 6 tasks and rephrasals of these tasks where they change only the noun, only the verb, both noun and verb and rewrite the entire task (human provided). This comes to a total of 77 instructions for all 6 tasks combined. An example is shown below and the full list of instructions can be found in the original paper.

1. Close drawer

2. Open drawer

3. Turn faucet left

4. Turn faucet right

5. Move black mug right

6. Move white mug down

Table 4: LOReL Example rephrasals for the instruction *"close drawer"*

| Seen | Unseen Verb | Unseen Noun | Unseen Verb + Noun | Human Provided |
|------|-------------|-------------|--------------------|----------------|
| close drawer | shut drawer | close container | shut container | push the drawer shut |

For the composition instructions, we took these evaluation instructions from the original paper and combined them to form 12 new composition instructions as shown below.

Table 5: LOReL Composition tasks

| Instructions |
| --- |
| open cabinet and move black mug right |
| pull the handle and move black mug down |
| shift white mug right |
| shift black mug down |
| shut drawer and turn faucet right |
| close cabinet and turn faucet left |
| turn faucet left and shift white mug down |
| rotate faucet right and close drawer |
| move white mug down and rotate faucet left |
| open drawer and turn faucet counterclockwise |
| slide the drawer closed and then shift white mug down |
| rotate faucet left and move white mug down |
| shift white mug down and shift black mug right |
| turn faucet right and open cabinet |
| move black mug right and turn faucet clockwise |

We included the instructions *"move white mug right"* and *"move black mug down"* as composition tasks here in the hope that we may have skills corresponding to colors like black and white or directions like right and down that can be composed to form these instructions but we did not observe such behaviour.

# D Training details

We plan to release our code on acceptance. Here we include all hyper-parameters we used. We implement our models in PyTorch. Our original flat baseline implementation borrows from Decision Transformer codebase which uses GPT2 to learn sequential behavior. However, we decided to start from scratch in order to implement LISA to make our code modular and easily support hierarchy. We use 1 layer Transformer networks for both the skill predictor and the policy network in our experiments for the main paper. We tried using large number of layers but found them to be too computationally expensive without significant performance improvements. In BabyAI and LOReL results we train all models for three seeds.

For BossLevel environment we use 50 skill codes, for other environments we used the settings detailed in the table below:

## D.1 LISA

Table 6: **LISA Hyperparameters**

| Hyperparameter | BabyAI | LORL |
| --- | --- | --- |
| Transformer Layers | 1 | 1 |
| Transformer Embedding Dim | 128 | 128 |
| Transformer Heads | 4 | 4 |
| Skill Code Dim | 16 | 16 |
| Number of Skills | 20 | 20 |
| Dropout | 0.1 | 0.1 |
| Batch Size | 128 | 128 |
| Policy Learning Rate | $1e-4$ | $1e-4$ |
| Skill Predictor Learning Rate | $1e-5$ | $1e-5$ |
| Language Model Learning Rate | $1e-6$ | $1e-6$ |
| VQ Loss Weight | 0.25 | 0.25 |
| Horizon | 10 | 10 |
| VQ EMA Update | 0.99 | 0.99 |
| Optimizer | Adam | Adam |

## D.2 Baselines

Table 7: **Flat Baseline Hyperparameters**

| Hyperparameter | BabyAI | LORL |
| --- | --- | --- |
| Transformer Layers | 2 | 2 |
| Transformer Embedding Dim | 128 | 128 |
| Transformer Heads | 4 | 4 |
| Dropout | 0.1 | 0.1 |
| Batch Size | 128 | 128 |
| Policy Learning Rate | $1e-4$ | $1e-4$ |
| Language Model Learning Rate | $1e-6$ | $1e-6$ |
| Optimizer | Adam | Adam |

**Hyperparameter choices.** The hyperparameters were chosen approximately by us based on our estimate of the number of skills in the dataset and the complexity of the dataset. For example, we use 20 skill codes for LOReL experiments while we use 50 skill codes for the BabyAI BossLevel experiments as mentioned above. These hyperparameter choices are by no means exhaustive nor optimal as our ablation study in section F.1 suggests that our choice of horizon is perhaps sub-optimal (we used H=10 for our experiments in Table 1 but appendix section F.1 suggests that H=5 is better). The ablation study also suggests that the performance is fairly stable for a reasonable range of hyperparameter choices removing the burden from the practitioner.

**Baseline Implementations.** For the original baseline for BabyAI, we used the code from the original repository. For the LOReL baseline, we used the numbers from the paper for LOReL Images. For LOReL States BC baseline, we implemented it based on the appendix section of the paper. We ran the LOReL planner from the original repository for the composition instructions.

### D.3 Ablations

As mentioned in the paper, all our ablations were performed on BabyAI BossLevel with 1k trajectories over a single seed for the sake of time. Unless otherwise specified, we use the following settings. We use a 1-layer, 4-head transformer for both the policy and the skill predictor. We use 50 options and a horizon of 10. We use a batch size of 128 and train for 2500 iterations. We use a learning rate of 1e-6 for the language model and 1e-4 for the other parameters of the model. We use 2500 warm-up steps for the DT policy. Training was done on Titan RTX GPUs.

### D.4 MI Calculation

We calculate mutual information between the language instructions ($L$) and the skill codes ($z$) by writing $MI(L, z) = H(L) - H(L|z)$. Our procedure for this is very simple and uses $\sim 10$ lines of code. Specifically, we first calculate $H(L|z)$ by assuming $p(L|z)$ to be gaussian $N(\mu, I)$ with unit variance, centered at the codebook vector $z$. We can calculate $\mu$ by finding the distance between the codebook vectors and the language vectors in the latent embedding space. We can similarly calculate the $H(L)$ by taking the expectation of $H(L|z)$ over all discrete codebook vectors.

```python
def MI(option_codes, lang_state_embeds):
    """Calculate entropy of options over each batch
    option_codes: [N, D]
    lang_state_embeds: [B, D]
    """
    with torch.no_grad():
        N, D = option_codes.shape
        lang_state_embeds = lang_state_embeds.reshape(-1, 1, D)

        embed = option_codes.t()
        flatten = rearrange(lang_state_embeds, '... d -> (...) d')

        distance = -(
            flatten.pow(2).sum(1, keepdim=True)
            - 2 * flatten @ embed
            + embed.pow(2).sum(0, keepdim=True)
        )
        cond_probs = torch.softmax(distance / 2, dim=1)

        # get marginal probabilities
        probs = cond_probs.mean(dim=0)
        entropy = (-torch.log2(probs) * probs).sum()

        # calculate conditional entropy with language
        # sum over options, and then take expectation over language
        cond_entropy = (-torch.log2(cond_probs) * cond_probs).sum(1).mean(0)
        return entropy - cond_entropy
```

## E   Detailed LOReL Sawyer results

We provide details results on the LOReL evaluation instructions below for LISA and the flat baseline in the same format as the original paper. The results are averaged over 10 runs. The time horizon used was 20 steps.

Table 8: **Task-wise success rates (in %) on LOReL Sawyer.**

| Task | Flat | LISA |
|---|---|---|
| close drawer | 10 | **100** |
| open drawer | **60** | 20 |
| turn faucet left | 0 | 0 |
| turn faucet right | 0 | **30** |
| move black mug right | 20 | **60** |
| move white mug down | 0 | **30** |

Table 9: **Rephrasal-wise success rates (in %) on LOReL Sawyer.**

| Rephrasal Type | Flat | LISA |
|---|---|---|
| seen | 15 | **40** |
| unseen noun | 13.33 | **33.33** |
| unseen verb | 28.33 | **30** |
| unseen noun+verb | 6.7 | **20** |
| human | 26.98 | **27.35** |

# F More experiments

## F.1 Ablation Studies

For the sake of time, all our ablations were performed with a 1-layer, 4-head transformer for the skill predictor and for the policy. All our ablations are on the BabyAI-BossLevel environment with 1k expert trajectories.

Our first experiment varies the horizon of the skills. The table below shows the results on BabyAI BossLevel for 4 different values of the horizon. We see that the method is fairly robust to the different choices of horizon unless we choose a very small horizon. For this case, we notice that a horizon of 5 performs best, but this could vary with different tasks.

Table 10: **Ablation on horizon.** We fixed the number of options to be 50 for these experiments

| Horizon | 1 | 5 | 10 | 50 |
|---|---|---|---|---|
| Success Rate (in %) | 32 | **52** | 47 | 47 |

We also tried varying the number of skills the skill-predictor can choose from and found that this hyperparameter is fairly robust as well unless we choose an extremely high or low value. We suspect using more skills worsens performance because it leads to a harder optimization problem and the options don't clearly correspond to specific language skills.

Table 11: **Ablation on number of options.** We fixed the horizon to be 10 for these experiments

| Number of Options | 10 | 20 | 50 | 100 |
|---|---|---|---|---|
| Success Rate (in %) | **47** | **47** | **47** | 43 |

## F.2 Can we leverage the interpretability of the skills produced by LISA for manual planning?

Since our skills are so distinct and interpretable, its tempting to try and manually plan over the skills based on the language tokens they encode. In the LOReL environment, using the same composition tasks as section above, we observe the word clouds of options and simply plan by running the fixed option code corresponding to a task for a certain horizon and then switch to the next option corresponding to the next task. This means we are using a manual (human) skill predictor as opposed to our trained skill predictor. While this doesn't work as well because skills are a function of both language and trajectory and we can only interpret the language part as humans, it still shows how interpretable our skills are as humans can simply observe the language tokens and plan over them to complete tasks. We show a successful example and a failure case below for the instruction *"close drawer and turn faucet right"* in figure 22. We first observe that the two skills we want to compose are $Z = 14$ and $Z = 2$ as shown by the word clouds. We then run skill 14 for 20 steps and skill 2 for 20 steps. In the failure case, the agent closes the drawer but then pulls it open again when trying to turn the faucet to the right.

## F.3 Do the skills learned transfer effectively to similar tasks?

We want to ask the question whether we can use the skills learned on one task as a initialization point for a similar task or even freeze the learned skills for the new task. To this end, we set up experiments where we trained LISA on the BabyAI GoTo task with 1k trajectories and tried to transfer the learned options to the

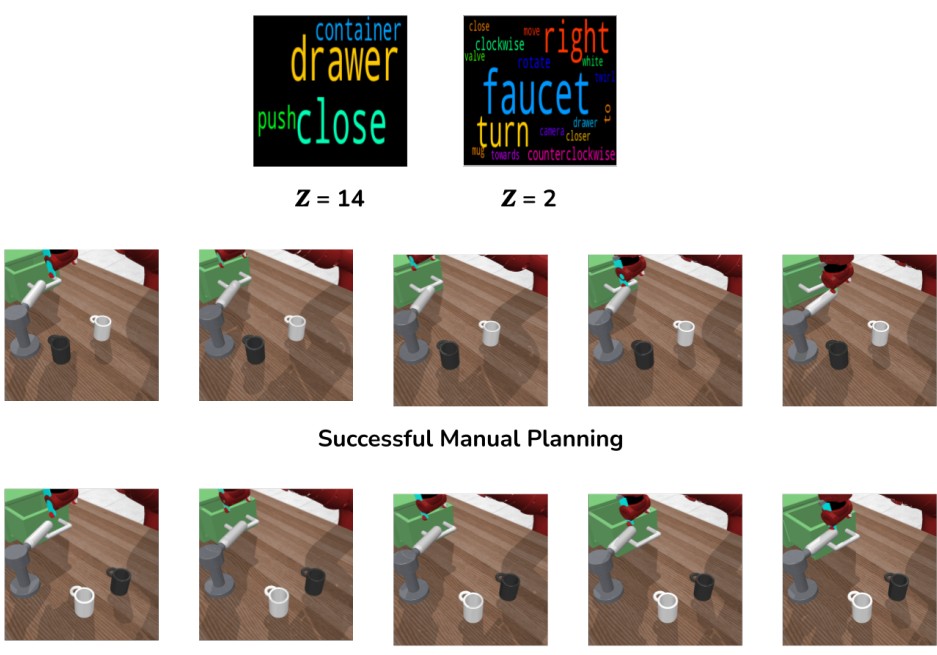

**Z = 14**          **Z = 2**

Successful Manual Planning

Unsuccessful Manual Planning

Figure 22: We show a successful manual planning and unsuccessful manual planning example for the instruction "close the drawer and turn the faucet right"

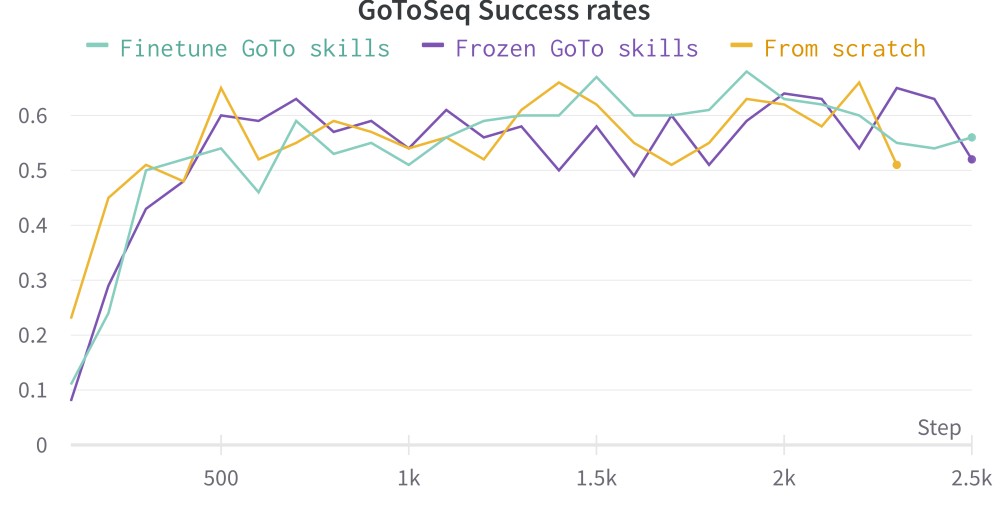

Figure 23: Transferring skills on the BabyAI GoToSeq environment

GoToSeq task with 1k trajectories. Similarly we trained LISA on GoToSeq with 1k trajectories and tried to transfer to BossLevel with 1k trajectories. The results are shown in figures 23 and 24 respectively.

As we can see from the GoToSeq experiment in figure 23, there is no major difference between the three methods. We notice that we can achieve good performance even by holding the learned option codes from GoTo frozen. This is because the skills in GoTo and GoToSeq are very similar except that GoToSeq composes these skills as tasks. We also notice that finetuning doesn't make a big difference – once again probably because the skills are similar for both environments.

In the BossLevel case in 24, however, we do notice that the frozen skills perform slightly worse than the other two methods. This is because the BossLevel contains more skills than those from GoToSeq. We also notice that the performance of finetuning and starting from scratch is nearly the same. This could be because the

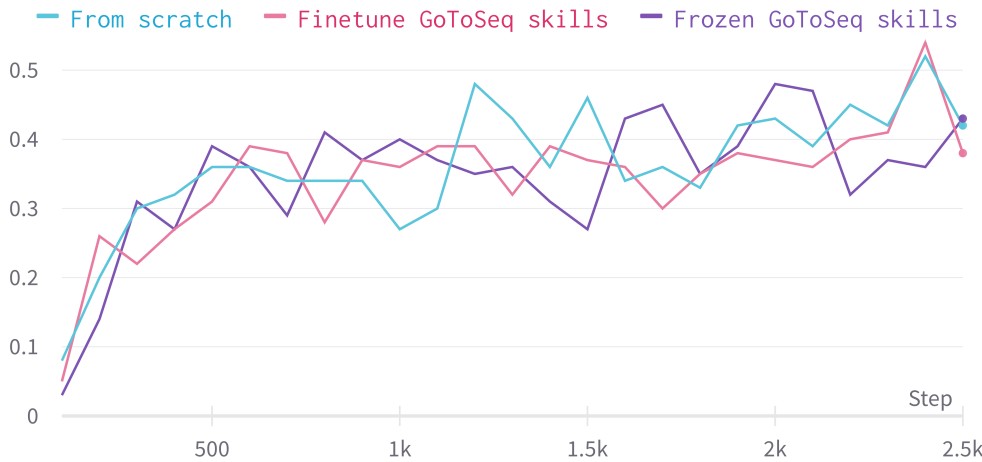

Figure 24: Transferring skills on the BabyAI BossLevel environment

meta-controller needs to adapt to use the new skills in the BossLevel environment anyway and there is no benefit from loading learned options here.

### F.4 Are the skill codes in LISA chosen because of environment affordances and not language?

A natural hypothesis for LISA choosing relevant skill codes for each task could be that they are chosen because of the environment affordances and not because of the language instructions. For example, LISA chooses the code corresponding to opening a drawer because the drawer is closed in the environment or because the robot arm is close to it when initialized. We systematically show that this is not the case. We initialize the LOReL environment with the drawer wide open and test all possible combinations of words meaning the same as "drawer" and "close".

- We issue the following language commands: "close drawer", "close container", "push drawer" and "push container". LISA is unable to solve the task "push container" but for the other three tasks, over 20 different environment seeds, LISA generates the skill code 14 (as shown in Figure 4), all 20 times for each instruction.

- We then issue the commands containing the object "drawer" with actions meaning different than "close": "rotate drawer", "rotate container", "move container down" and "move drawer down", "open drawer", "open container", "pull drawer", "pull container". We observe that none of these instructions used skill code 14 even once out of 20 different runs of each instruction.

- Next, we ran the commands containing the object different from "drawer" and actions same as "close": "close mug", "close faucet", "push mug" and "push faucet" and none of these instructions use the skill code 14 even once out of 20 different runs of each instruction.

These experiments indicate that the skill code 14 clearly is generated by LISA only when the "close drawer" instruction or a synonym is given in the language instruction. This clearly highlights that the codes are highly correlated to language and are not just being chosen based on the affordances of the environment.

### F.5 State-based skill-predictor

We have already spoken in section 3.2 about the fact that our method using two transformers doesn't necessarily mean its more compute-heavy than the non-hierarchical counterpart. But to test whether we really need two transformers, we perform an ablation study that replaces the skill predictor to be just a state-based selector MLP as opposed to a trajectory-based transformer. Our results show that the performance is only slightly worse when using a state-based skill predictor in this case, but once again this may not be the case in more complex environments. However, we notice that the skills collapsed in this case and the model tends to use fewer skills than normal as shown in figure 25. This is expected because the skill predictor is now predicting skills with much less information.

### F.6 Continuous skill codes

We also compare to the non-quantized counterpart where we learn skills from a continuous distribution as opposed to a categorical distribution. We expect this to perform better because the skill predictor has access to a larger number of skill codes to choose from and this is what we observe in table 13. However, this comes

Table 12: **Comparing state-based MLP skill predictor vs trajectory-based transformer skill predictor.** We fixed the number of options to be 50 and horizon as 10 for these experiments

| Skill Predictor Architecture | Success Rate |
|---|---|
| State-based MLP | 46% |
| Trajectory-based Transformer | **47%** |

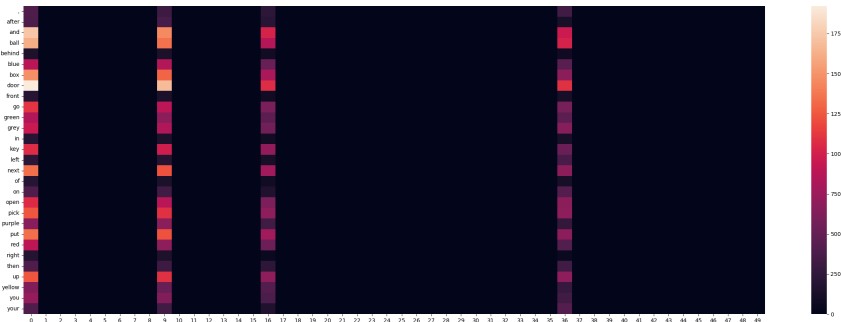

Figure 25: State-based skill predictor heat map shows that the model tends to use fewer options compared to figure 8

at the price of interpretability and its harder to interpret and choose continuous skill codes than discrete codes. We also observe that on the LOReL with states environment, using discrete codes performs better than using continuous codes (table 14). This could be because learning a multi-modal policy with discrete skills is an easier optimization problem than learning one with continuous skills (see the end of section 4.3).

Table 13: **Ablation on Quantization on BabyAI BossLevel.** We fix number of options to 50 and horizon to 10

| Skill codes | Success Rate (in %) |
|---|---|
| Continuous | **51** |
| Discrete | 47 |

Table 14: **Ablation on Quantization on LOReL with states on seen tasks.** We fix number of options to 20 and horizon to 10

| Skill codes | Success Rate (in %) |
|---|---|
| Continuous | 60.0 |
| Discrete | **66.7** |