# OpenReview forum: "LISA: Learning Interpretable Skill Abstractions from Language"
_NeurIPS.cc/2022/Conference — NeurIPS 2022 Accept_

### Official Review · Reviewer_f8PZ · 2022-07-04

**Rating:** 6
**Confidence:** 4
**Soundness:** 3 good
**Presentation:** 3 good
**Contribution:** 3 good

**Summary:**

This paper proposes a hierarchical model for instruction-conditioned imitation learning (instruction following). A skill predictor network conditions on instructions and observations to predict discrete skill codes, and a policy network conditions on these skill codes to carry out actions. Discrete skill codes are learned using vector quantization. The paper evaluates on BabyAI and LOReL imitation learning tasks, finding that (1) the method outperforms a non-hierarchical imitation learning baseline model in low-resource training settings and when carrying out complex compositional tasks and (2) discrete skills induced are correlated with words in the language.

**Questions:**

Q1) Could more details be clarified on the difference between instructions seen in training and in evaluation for both BabyAI and LOReL, especially in the "new composition" settings?

Q2) Is 4.7, is the time horizon increased at inference time only or just at training time? It's also unclear to me why H would need to be increased, since if I understand correctly it's the length of time each skill is invoked for and it seems that while the task contains more skills in this setting, skills themselves need not be longer.


**Limitations:**

I appreciated the discussion of the option-related limitations: needing to fix the horizon, and number of options. It would help to more clearly outline how the values for these were chosen in this work.

*Suggestions*

- I think it would really strengthen the paper to either

(1) extend the "manual planning" experiments given in the Appendix to show that control codes can be intervened on to make the models carry out behavior in novel contexts where the behaviors weren't seen in training or

(2) show that the models can carry out unseen compositions of sub-tasks (i.e. similar to the compositional generalization setting explored in Corona et al.)

- The claims in 294 and 302-305 about sub-task reuse, and skill codes capturing task variation were pretty abstract and it would really strengthen the paper if some concrete evidence for them could be presented.

*Minor comments*

- line 105: Hu et al. does learn interpretable skills - language descriptions of skills are produced by their instructor model.
- 145: the description here could be a bit more rigorous. Is there mean to be a latent partitioning of each trajectory into steps for each sub-task, or can steps be associated with multiple sub-tasks in some soft way?
- 171: it would help to give more details on how H is set, perhaps with references to the values & ablations in the appendix.
- 181: this equation above doesn't describe the learning process for the cluster centers.
- 198: this is not obvious to me; a citation would help.
- 198: fragmented sentences
- 242: "off" -> "of"



**Strengths And Weaknesses:**

*Strengths*

S1) The approach in the paper is well-motivated and is likely to be of interest both to the sequential decision making and language grounding communities. Most past work on hierarchical language-conditioned imitation learning has relied on supervising the intermediate layer of the hierarchy, so an end-to-end approach is exciting.

S2) The method is generally well-described, although some details were only clear from the appendix.

S3) I appreciate that the paper evaluated on two different datasets, and presented pretty thorough ablations and qualitative analysis (in the appendix).

*Weaknesses*

W1) The main novelty of the paper is the learning of discrete skills from language & demonstrations in an end-to-end fashion -- without the discrete skill bottleneck, it seems that the LISA approach would be an end-to-end neural instruction following model of the type very common in work on language grounding (e.g. Mei et al 2016, Anderson et al. 2018), and presented as an ablation in appendix F.2. ~~But the paper doesn't currently have enough evidence to convince me that these discrete skills are useful:~~

W1a) ~~While LISA does outperform a baseline approach (the "flat" model), this baseline has a pretty different architecture and initialization (in particular, LISA uses a pre-trained DistillBERT text encoder while the baseline model is trained from scratch) and so I think is much less suitable than as a baseline than the ablation in appendix F.2, which performs comparably to LISA. So I don't think it's reasonable to claim performance improvements from the LISA approach based on the current results.~~

_Update after response:_ Thanks to the authors for the baseline clarifications; the baseline is much more comparable to LISA than I'd thought.

W1b) It's unclear to me that the use of discrete codes adds interpretability. The main interpretation method (word clouds and heatmaps) maps these skills to words in the existing instructions. Similar methods can also be applied to continuous representations (e.g. Andreas et al., Translating Neuralese). In the skill code fixing example (4.5), it seems plausible that the method is just carrying out "close the drawer" given the affordance of the environment: the arm is located next to an open drawer. See suggestions, in the limitations section, below for suggestions on this.

_Update after response:_ I appreciate the author's arguments about the easy of interpretability, and the additional experiments showing that skill codes are being used and it's not just environment affordances.

W2) Although not a crucial weakness, since this paper does contribute beyond them, some relevant works are

Sharma et al. Skill Induction and Planning with Latent Language ACL 2022

Corona et al. Modular Networks for Compositional Instruction Following NAACL 2021

W3) Some of the experimental descriptions could be clearer, in particular whether the paper is interested in evaluating generalization to instructions unseen at training time; see questions below.

---

> ### Author Response · Authors · 2022-08-02
> **Author Response**
>
> Thanks for your time and effort in the in-depth assessment of our work. Great suggestions and comments. We were delighted that you found our approach well-motivated and an exciting line of work.  We do our best to address your concerns below and note that most may be allayed with improved clarity and details, which we will add to our revision. We would love to answer follow-up questions if any concerns are not addressed.
>
> **Comparisons with Flat Baseline.** For the sake of brevity in our submission, we didn’t elaborate on the details of the implementation of the flat baseline, and we will expand on them to add much more clarity in our next version. In particular, in our experiments, the flat baseline uses the same pre-trained DistillBERT text encoder model as LISA for dealing with natural language input. Furthermore, the flat baseline and LISA's skill predictor and policy share the same building blocks and are all implemented as Causal Transformer models with the same initialization schemes.
> Finally, to make LISA’s comparisons fair to the flat baseline we take care in ensuring the number of parameters are roughly the same for both. For e.g. the flat baseline uses a 2 layer Transformer network, and LISA’s skill predictor and policy use a 1 layer Transformer network each (shown in Table 5 and 6 in the supplementary). We will rephrase our implementation details section in the main text to include the above important details.
>
> **Interpretability of discrete skill codes.** Thanks for suggesting the ‘Translating Neuralese’ work on adding interpretability to continuous representations. We were not aware of it and we will add it to our related work. Nevertheless, adding interpretability to continuous representations requires workarounds, like learning conditional probability distributions, which can introduce extra complications. Whereas for discrete codes, interpretability can be achieved directly and allows use of off-the-shelf techniques like correlation maps, and word clouds.
>
> To highlight the interpretability strength of the learned discrete codes and the code reuse in LISA, we run some composition experiments (these are all unseen instructions) and present some GIFs and word clouds in the updated supplementary material in the *LISA Media* folder.
>
> These are similar to the manual planning experiments requested by the reviewer except all the planning here is done by LISA. We can clearly see the skills corresponding to the word cloud behaviors for each instruction.
>
> **Are the skill codes chosen because of environment affordances and not language?** This is a great question and something we hadn’t considered! We ran a simple experiment to test this hypothesis given in a separate thread below to prove that this is not indeed the case.
>
> **Details on instructions seen in training and evaluation.**
> Thanks for pointing this out! In our initial draft, we only had 5 unseen and 10 seen instructions in our results for Table 2. We have since run more experiments with **15 unseen instructions** and the results are provided in our response to Reviewer txnz. We also provide detailed statistics about the % of unseen instructions in the BabyAI evaluation.
>
> **Time horizon at inference.** Sorry for the confusion, H is not increased between train and test time, and here we mean that the overall timesteps for policy execution are increased to allow the policy to finish longer composition tasks.  In the LORL paper, at test time the agent is allowed to take 20 steps in the environment to achieve the task but this was for single goal tasks. Since most of our composition tasks have two goals, we gave the model twice the amount of time i.e. 40 steps.
>
> **Choice of hyperparameters.** We apologize for not elaborating more on the choices of hyperparameters in the main paper. The hyperparameters were chosen rather arbitrarily by us based on our estimate of the number of skills in the dataset and the complexity of the dataset. For example, we use 20 skill codes for LOReL experiments while we use 50 skill codes for the BabyAI BossLevel experiments as given in Appendix section D1. These hyperparameter choices are by no means exhaustive nor optimal as our ablation study in section F1 suggests that our choice of horizon is perhaps sub-optimal (we used H=10 for our experiments in Table 1 but appendix section F1 suggests that H=5 is better). The ablation study also suggests that the performance is fairly stable for a reasonable range of hyperparameter choices.
>
> **Related work.** Thanks for pointing out the relevant works. We were not aware of them, but they are very interesting and indeed in a similar vein and we will discuss them in related works.
>
> **Notation Fixes.** We will improve the clarity and try to incorporate the suggested fixes in the next version.

---

> > ### Author Response · Authors · 2022-08-02
> > **Are the skill codes chosen because of environment affordances and not language?**
> >
> > To test this hypothesis we run the following experiment:
> >
> > We initialize the LORL environment with the drawer wide open and test all possible combinations of words meaning the same as "drawer" and "close"
> >
> > 1) We issue the following language commands:
> > “close drawer”, “close container”, “push drawer” and “push container”. LISA is unable to solve the task “push container” but for the other three tasks, over 20 different environment seeds, LISA generates the skill code 14, all 20 times for each instruction ([word cloud link for skill code 14](https://ibb.co/MGpVxSz))
> >
> > 2)  We then issue the commands containing the object "drawer" with actions meaning different than "close": “rotate drawer”, “rotate container”,  “move container down” and “move drawer down”, "open drawer", "open container", "pull drawer", "pull container". We observe that none of these instructions used skill code 14 even once out of 20 different runs of each instruction.
> >
> > 3) Next, we ran the commands containing the object different from "drawer" and actions same as "close": “close mug”, “close faucet”,  “push mug” and “push faucet” and none of these instructions use the skill code 14 even once out of 20 different runs of each instruction.
> >
> > These experiments indicate that the skill code 14 clearly is generated by LISA only when the "close drawer" instruction or a synonym is given in the language instruction. This clearly highlights that the codes are highly correlated to language and are not just being chosen based on the affordances of the environment.

---

> > ### Comment · Reviewer_f8PZ · 2022-08-07
> > **Post-response update**
> >
> > Thanks to the authors for the thorough response! It, and the responses to the other reviewers as well, addressed all of my main concerns. I've raised my score to a 6 (from a 4):
> >
> > *Comparison to flat baseline*
> > Thank you for these clarifications, which, in combination with the response to k4fJ, addressed my concerns. Please do add these details in future versions of the paper.
> >
> > *Interpretability of discrete skill codes*
> > These are great points -- I appreciate that the proposed technique allows simpler off-the-shelf interpretability techniques such as word-clouds.
> >
> > *Skill code affordance experiments*
> > This was well-done and well-described, and I found it convincing. I think it would be a great addition to the appendix of the final version of the paper.
> >
> > *Unseen instructions*
> > I also appreciated the experiments described in the response to txnz, which helped convince me about the generalization ability of the approach.

---

### Official Review · Reviewer_k4fJ · 2022-07-08

**Rating:** 6
**Confidence:** 4
**Soundness:** 3 good
**Presentation:** 2 fair
**Contribution:** 3 good

**Summary:**

***Post rebuttal update: Raising score from 5 to 6***


This work proposes a method to learn a hierarchical instruction following agent policy from demonstrations.The policy is composed of a skill predictor which plays the role of a high-level controller, and a low level controller predicts actions conditioned on the skill. The skills are trained from scratch with a vector quantization approach; the skills are represented by codes in a codebook and the skill predictor chooses the skill to execute based on a language instruction and a history of observations. The proposed method is shown to perform well on BabyAI and Lorel Sawyer benchmarks and outperforms baseline non-hierarchical policies on long horizon tasks, especially when supervision is limited. In addition, the authors show that the learned skill vectors are interpretable and correspond to meaningful primitive skills.

**Questions:**

* How is the MI computed in Fig 6?
* Have you considered alternative approaches where you simply segment the trajectories and cluster them to identify salient skills? I believe this could be a strong baseline. Can you provide some intuition to why the vector quantization approach would be preferred over this approach?

Presentation
* 132: “task can be a union of other tasks” - this is not clear to me
* 136: “sub-goals for the task are encoded within this single instruction” - didn’t understand this
* 217: “finetune the language encoder to the vocabulary” - what does this mean?
* There are mentions of ‘GIF’ in the paper. Good to rephrase.


**Limitations:**

Seems adequate.

**Strengths And Weaknesses:**

Strengths
* Learning reusable skills from data is an important problem with potential for better generalization, interpretability and controllability.
* Proposed approach outperforms baseline approaches when supervision is limited.
* Authors show that the learned skills are interpretable, meaningful and lead to better generalization on long-horizon tasks.

Weaknesses
* The precise source of performance is hard to pinpoint. Does LISA work better than the flat baseline because it has more parameters? Were hyperparameters of baseline approaches carefully tuned? The authors can try to better isolate the source of performance.
* The proposed approach has many hyperparameters to be tuned such as codebook size, horizon, etc. I could not find discussion about how these parameters were chosen.
* The fixed horizon assumption is a limitation. Have the authors examined how the choice of this parameter affects the results?

The proposed approach is interesting and shows positive results on two benchmarks. But the approach has key limitations which needs to be better addressed in the paper. This includes discussion about hyperparameter choices and how they affect the performance and more clarity on how the skills are interpreted. Based on the quantitative results presented on the paper alone it is hard to gauge the proposed approach due to confounding factors such as hyperparameter optimization and parameter count. Addressing these issues can significantly improve the paper.

I found the discussion in sec 4.7 interesting. The proposed approach can potentially work very well on compositional generalization. Expanding the scope of the analysis (Eg. defining a clear evaluation setup, experimenting on a large number of tasks, etc.) in sec 4.7 can significantly strengthen the paper and become a core contribution.

Other
* I couldn’t find enough details in the paper to understand how the correlation between skills and words is computed and how it is visualized.
* Algorithm 1: line 12 needs to be described better
* Plots are hard to visualize due to small font sizes (fig 4, fig 6). Also, there are layout issues that need to be fixed (Unclear why Fig 6 appears in p8)

---

> ### Author Response · Authors · 2022-08-02
> **Author Response**
>
> Thank you for your time and effort in the in-depth assessment of our work. We are encouraged you found our work interesting and the learning of reusable skills important.  We do our best to address your concerns below and would love to answer follow-up questions if any concerns are not addressed. Thanks again!
>
> **Parameter count of LISA vs Flat Baseline.** Great question! We were wary of this when we conducted our experiments as well.  Since LISA has two transformers as opposed to just one in the flat baseline we had to be careful about ensuring that the baseline and our method had a similar number of total parameters. To this end, the flat baseline uses a 2-layer Transformer network, and LISA’s skill predictor and policy use a 1-layer Transformer network each. We also ensured that the embedding dimension and the number of heads in each layer were exactly the same in both LISA and the flat baseline. Details of this are provided in appendix sections D1 and D2. In fact, one could argue that LISA has less representation power because the policy transformer can only attend to the last H steps while the flat baseline can attend to the entire trajectory which is what makes it an extremely strong baseline.
> We also tuned the other hyperparameters like learning rate, dropout and batch size for both LISA and the flat baseline and found that the same values worked well for both models. It is important to note that the flat baseline is extremely powerful and we verified this by training it on the different datasets and it achieved better accuracies than the original papers by training for much less time. We also find our flat baseline to be SOTA on the BabyAI-Bosslevel task on similar data regimes compared to prior methods [1].
>
> **Choice of hyperparameters.** We apologize for not elaborating more on the choices of hyperparameters in the main paper. The hyperparameters were chosen rather arbitrarily by us based on our estimate of the number of skills in the dataset and the complexity of the dataset. For example, we use 20 skill codes for LOReL experiments while we use 50 skill codes for the BabyAI BossLevel experiments as given in Appendix section D1. These hyperparameter choices are by no means exhaustive nor optimal as our ablation study in section F1 suggests that our choice of horizon is perhaps sub-optimal (we used H=10 for our experiments in Table 1 but appendix section F1 suggests that H=5 is better). The ablation study also suggests that the performance is fairly stable for a reasonable range of hyperparameter choices.
>
> **Fixed, common horizon for all the codes.** It is indeed a limitation of LISA that the horizon of each skill is uniform and has to be chosen ahead of time and we hope to alleviate this in future work. In this paper, to keep things simple and to focus our study on the nature of skills learned and their interpretability and efficacy in the low data regime, we did not experiment with trying to learn the horizons along with the skill codes. We do conduct an ablation study on the choice H in Table 9 of the Appendix, finding different values to work fairly well. Nevertheless, learning of the horizons can be important to learn even better skills and we hope to address it in future.
>
> **Evaluation setup and scope of analysis.**
> Thanks for pointing this out - we'll elaborate on this in the main paper and discuss our expanded scope in the additional comment below.
>
> **Correlation calculations and the visualizations.** We give details on this in Appendix section B1. We apologize for not pointing to this in the main paper, for the sake of brevity, and we will fix this in the next version.
>
> **MI Calculation.** We calculate mutual information between the language instructions ($L$) and the skill codes ($z$) by writing $MI(L, z) = H(L) - H(L|z)$. Our procedure for this is very simple and uses less than 10 lines of code and we will add a code snippet on MI calculation to our appendix in our revision. Specifically, we first calculate $H(L|z)$ by assuming  $p(L|z)$ to be gaussian $\mathcal{N}(\mu, I)$ with unit variance, centered at the codebook vector $z$. We can calculate $\mu$ by finding the distance between the codebook vectors and the language vectors in the latent embedding space. We can similarly calculate the $H(L)$ by taking the expectation of $H(L|z)$ over all discrete codebook vectors.
>
> **Clustering trajectories baseline.** This is a great suggestion and we address it in a separate comment below!
>
> **Notation Fixes.** We will improve the clarity and incorporate the suggested fixes in the next version.
>
> [1] Hejna et al - Improving Long-Horizon Imitation Through Language Prediction

---

> > ### Author Response · Authors · 2022-08-02
> > **Evaluation setup and scope of analysis**
> >
> > In our response to Reviewer txnz, we have clearly described the evaluation statistics of different tasks in terms of the number of unseen instructions at evaluation time. We have also provided fresh results for Table 2 with completely unseen instructions.
> > It is also important to note that in the composition analysis setting, we compare LISA with a very strong Lorel planner baseline, which uses a pre-trained visual dynamics model that is pre-trained on 1M frames whereas LISA is trained from scratch.
> > Moreover, as already mentioned, the flat baseline and LISA have roughly the same number of parameters and are almost identical apart from the vector quantization component. This lends more credence to the hypothesis in our work that the skill quantization is the source of the improved performance.
> > In terms of the number of composition instructions, unfortunately, we are limited here by the number of diverse subtasks present in the Lorl dataset which is roughly 12. We handcrafted the composition dataset to include 15 new instructions such that it covers a wide range of scenarios that are not seen in the given table-top environment with a limited number of objects to manipulate. Nevertheless, to reduce bias in our results, we ran each instruction 10 times with 3 different seeds and initializations of the environment.
> >
> > If you have any other specific questions about our evaluation setup, we’re more than happy to answer them!

---

> > > ### Comment · Reviewer_k4fJ · 2022-08-09
> > > **Thanks for the response, raising my score from 5 to 6**
> > >
> > > I thank the authors for addressing my review comments. The rebuttal addressed some of my concerns about the experimental setup and the interpretation of results, and I am raising my score from 5 to 6. But at the same time, I encourage the authors to carefully address the concerns about hyperparameters in their revision.

---

> > ### Author Response · Authors · 2022-08-02
> > **Clustering trajectories baseline**
> >
> > We first give an explanation of why VQ works here, and then construct a simple clustering-based baseline to compare against it. In our case, the VQ approach can be seen as taking the concatenated language-state inputs and projecting it into a learnt embedding space. VQ here simply learns K embedding vectors that act as K cluster centers for the projected input vectors in this embedding space, and allows for differentiability, enabling learning through backpropagation. This is similar to propotypical methods used for few-shot learning and allows for deep differentiation clustering, thus giving an intuition of why it works.
> >
> > Now to compare against this approach, we construct a simple unsupervised learning baseline that clusters trajectories using K-means. Specifically, in the BabyAI-BossLevel environment using 1K training trajectories, we take all concatenated language-state vectors for all trajectories in the dataset and cluster them using K-means and use the assigned cluster centers as the skill codes. We then learn a policy using these skill codes to measure their efficacy and report results below over three seeds:
> >
> > **LISA (Ours):   49.1± 2.4**
> >
> > K-means skills:  20.2 ± 5.2
> >
> > Thus, we see that using the simple K-means baseline skills is insufficient to learn a good policy to solve the BabyAI BossLevel task, as the skills are not representative enough of the language instructions. A reason for this is that language and state vectors lie in different embedding spaces, and K-means based on euclidean distance is not optimal on the concatenated vectors.

---

### Official Review · Reviewer_txnz · 2022-07-11

**Rating:** 6
**Confidence:** 3
**Soundness:** 3 good
**Presentation:** 2 fair
**Contribution:** 2 fair

**Summary:**

The submission presents a method, LISA, a hierarchical imitation learning framework that train a language-conditioned agent with an end-to-end fashion. LISA learns a skill abstraction through a skill predictor module and vector quantization conditioned on language input and observations, a policy module is learned conditioned on the skill abstraction. LISA is trained with behavior-cloning and vector quantization objectives. LISA's performance is evaluated on two tasks, BabyAI, LOReL; some baselines are created by authors and from original task paper. Comparing with baselines, the results demonstrate good sample efficiency, possibility for perform unseen composition task and good interpretability from learned skill code.

**Questions:**

1. line 222: I am confused why use a causal mask for hiding future observation since skill predictor only takes previous states and language as input according to line 164.
2.  line 257 and line 267 describing generalization and compositionally related claims, but how likely the evaluation set is unseen is not quantified.
3.  The composition task dataset appears to be small, is a larger version available at this time?

**Limitations:**

The hyperparameter limitations are included in the paper. The interpretation ability of skill code is limited and relies on problem setting.

**Strengths And Weaknesses:**

### Originality
It is a recombination of existing techniques, where vector quantization is used in representation learning and behavior cloning is also not new. Nonetheless, it is the first work that use a combination that learn a quantized abstract from language and state, and develop a policy based on the produced skill abstract.
### Quality
The technical details are provided in the writing or references are given. Most experiment result and analysis are well supported. I believe the claims made in the contribution part are mostly fulfilled. The *weakness* of the paper are: the number of skill code is task dependent, the horizon of each skill has to be fixed, and chosen ahead; (the weakness aforementioned are mentioned in the writing). The conclusion related to compositionally is less convincing for me as I posted in question section below. The skill codes interpretation make more sense for LOReL than BabyAI due to larger of vocabulary size, thus, there are limitations on how we could use this interpretation.
### Clarity
The writing is mostly clear, but there are errors in table name (eg Table 10 referred multiple times but it should be table 1). Also table and figure are often too far from the description (Fig 3, Tab 1); line 173, does the policy module take input of both state and skill code or only skill code? The two options for LOReL task in table 1 is not explained in the paper.
### Significance
LISA presents a new hierarchical imitation learning framework that utilize vector quantization to learn skill abstraction and further build a policy module on top of skill abstractions. LISA is sample efficient and is useful for low-data regime. The learned skill codes are interpretable on limited conditions.

---

> ### Author Response · Authors · 2022-08-02
> **Author Response**
>
> Thank you for your time and effort. Great comments and suggestions. We are encouraged that you noticed LISA to be the first work to learn discrete abstractions from language for hierarchical learning in a control setting. This is something we are very excited about and the potential doors it opens in learning of controllable policies, not just in IL but also for RL in the future. We do our best to address your concerns and put clarifications below. We would love to answer follow-up questions if any concerns are not addressed.
>
> **Causal Masking.** You are correct that the skill predictor only takes previous states and language as input and this is exactly what happens at evaluation. The use of causal masking is an implementation detail to make training more efficient for the Transformer model similar to how GPT-2 and Decision Transformer works. At train time, we use the causal mask in the Transformer model to predict all the outputs simultaneously instead of predicting only one output at a time and feeding it in at the next step.
>
> **Unseen eval statistics.**
>
> _BabyAI_: To give a sense of the % of unseen instructions for BabyAI when we evaluate on the gym environment, we take the different BabyAI environments and report the % of unseen language instructions seen at eval time for different training data regimes in the table below. For each statistic, we sample 10,000 random instructions from the environment and check how many are unseen in the training dataset used, repeated over 3 different seeds.
>
> | Environment      | 1K trajs| 10k trajs | 100k trajs |
> | ----------- | ----------- | ----------- | ----------- |
> | GoToSeq      | 76%       | 63.8% | 48% |
> | SynthSeq   | 76.3%      | 66% | 50.7% |
> | BossLevel   | 77.3%      | 66.9% | 51.3% |
>
> This should give a sense of how many unseen instructions we can encounter when we train with very limited training data.
>
> _LOREL_: Table 2 in the paper had 5 unseen instructions and 10 seen instructions out of 15. Since our first draft, we re-ran that experiment with **all 15 unseen instructions** and we produce the results below:
>
> | Model      | Success  |
> | ----------- | ----------- |
> | Flat      |   13.33 $\pm$ 1.25%     |
> | LOREL Planner   | 18.18 $\pm$ 1.8% |
> | **LISA (Ours)**   | **20.89 $\pm$ 0.63 %**  |
>
> Once again, these results are over 10 runs of each instruction and 3 different seeds (effectively 30 different runs of each instruction). These results are pretty similar to the ones reported in the main paper as well.
>
> **Composition dataset size.** Unfortunately, we are limited here by the number of diverse subtasks present in the LOREL dataset which is roughly 12. We handcrafted the composition dataset to include 15 new instructions such that it covers a wide range of scenarios that are not seen in the given table-top environment with a limited number of objects to manipulate. Nevertheless, to reduce bias in our results, we ran each instruction 10 times with 3 different seeds and initializations of the environment.
>
> **Hyperparameter Choices.** In our experiments in Table 1, we chose rough values for the horizon and the number of options without careful tuning, simply based on the nature of the dataset and our estimate of the number of skills required. Our ablation study in Appendix section F1 suggests our performance is fairly robust within reasonable range of the codebook size and horizon choices, without adding much burden to a practitioner.
>
> **Interpretation ability.** In our generations of word clouds and correlation heat maps, we only know the entire language instruction and the corresponding skill codes, without knowing the exact mapping from word tokens to skill codes. To resolve this, we simply correspond each skill to each token in an instruction-skill code pair which introduces a lot of noise. This is explained in detail in Appendix B1.
> In future, we plan to investigate better interpretability techniques like visualizing attention maps between the skill codes and language tokens. Nevertheless, the discrete learning afforded by LISA can be seen as a first step towards learning interpretable and controllable skills that could transfer to RL settings as well.
>
> **Notation Fixes.** We will improve the clarity and incorporate the suggested fixes in the next version. In line 173, the policy module does take input of both state and skill code and we will fix this typo.
>
> **Table 1 Fixes.** We apologize for not explaining this clearly in the paper. The LOReL dataset comes with trajectories containing partially-observed image observations as well as fully-observed internal states of the robot and environment. We evaluate using both the settings for Table 1 results.

---

> > ### Comment · Reviewer_txnz · 2022-08-09
> > **Response**
> >
> > I appreciate authors for clarifications and actively addressing my questions.
> >
> > I found the provided statistics about unseen data and the additional unseen data experiments convincing.
> >
> > Also, I understood better about the performance contribution of learning skill code with VQ and interpretability of the learned skill code based on the response to k4fJ and f8PZ.
> >
> > I raised my score from 5 to 6.

---

### Official Review · Reviewer_qGvo · 2022-07-15

**Rating:** 6
**Confidence:** 4
**Soundness:** 3 good
**Presentation:** 3 good
**Contribution:** 3 good

**Summary:**

The paper proposes a hierarchical imitation learning where language instruction is used as input and converted to a latent code. This latent code is then used to condition the policy and guide the policy to the target.

**Questions:**

The approach seems to be a simple extension to various works in unsupervised skill discovery methods such as DIAYN, RVIC, etc. The difference being here the latent codes are not randomly initialized but instead are learned through VQVAE. I'm curious though how would LISA perform if all the instructions and the resulting codes are given at once and not sampled every H steps. Is it the sampling every H steps that's important to the resulting behavior?

Is the choice of using VQVAE to convert to discrete codes arbitrary? How about using approximate backpropagation techniques such as Gumbel-Softmax? Would this change the result?

Did the authors look into the effective number of skills used? This helps in highlighting that all codes in the codebook are indeed useful and has some relationship with some linguistic object.

**Limitations:**

The authors discuss the limitations of the work in the last section. Besides tuning hyperparameters, is there any limitation on the length of the language instruction used? How does the approach scale when we increase the number of skills in the codebook for the same task?

**Strengths And Weaknesses:**

The paper is nicely written and the main arguments are clearly explained. The figures although are a bit harder to discern due to the smaller size. The proposed approach, LISA, is one of the first imitation learning methods that employs learning low-level skills from a set of instructions.

The results show that the agents are able to outperform the baselines and the word clouds depict the different skills associated with different latent variables.

---

> ### Author Response · Authors · 2022-08-02
> **Author Response**
>
> Thanks for your time and effort. Nice suggestions and comments. We are encouraged that you found our paper well written and the arguments clear. We do our best to address your concerns and provide clarifications below. We would love to answer follow-up questions if any concerns are not addressed.
>
> **Relation to unsupervised skill discovery.** We agree LISA shares similarity with unsupervised learning methods like DIAYN, RVIC, nevertheless, these works learn unconditioned skills that maximize the state space coverage of the agent. For e.g. DIAYN’s objective is to maximize the mutual information (MI) between the skills and visited states. Here, what skills are learnt is not controllable and the policy behavior output corresponding to a skill may not be desirable/useful for downstream tasks.
> LISA in contrast is an end-to-end method for learning controllable skills conditioned on natural language input via imitation learning. This solves the two issues with the above methods: 1) learning controllable skills, and 2) getting desired policy behavior that is similar to an expert agent. Our objective takes inspiration from DIAYN but does this for language, i.e. maximizing the MI between the skills and natural language input.
>
> **Sampling of LISA skills.** For choosing a new skill, we need to take into account the current state of the agent which is not known ahead of time, and prevents predicting all skill codes at once. Thus, this presents a tradeoff for the choice of the skill sampling horizon H. If H is chosen to be +inf, i.e. only sampling a single skill code using the initial state, then we will be using the same single skill for the entire policy execution. Similarly, if H is chosen to be 1, then we will be sampling a new skill for every policy step. In both cases, the learnt skills will fail to act as Options [1], i.e. a hierarchical temporal abstraction over the actions.
> The choice of the horizon or a termination-condition for a skill is a well-studied issue in the Option literature, see [2, 3], and H here controls the granularity of the learnt skills to decompose the agent behavior.
>
> **Learning using Gumbel-Softmax.** We agree that Gumbel-Softmax can be a viable option for learning discrete skills. In fact, in our preliminary experiments, we tried using Gumbel-Softmax but found it to be hard to train in a stable manner and our skills often collapsed to a single code without any diversity. In contrast, we found that using VQ led to stable training and learning of diverse skills. This is well justified theoretically as the VQ loss used to train the discrete skill predictor matches LISA’s objective of maximizing the MI between the skills and the language (See Sec 3.2, Avoiding language reconstruction)
>
> **Effective number of skills.** We analyzed the effective number of skills used by visualizing word-skill heatmaps for different task settings. For example in Fig. 4, we can see that the skills are sparsely activated for different words, and 10 out of the total 20 skill codes are being effectively used. Similarly, in Fig.7 in the supplementary we see that skills are very densely activated in the skill heatmap for BabyAI BossLevel task.
>
> **Length of the language instruction.** The length of the language input is purely limited by the language model used in our case. In our experiments, we use DistilBERT as the language model which allows using upto 512 tokens in the input. This is orthogonal to our method as we can simply replace DistilBERT with a language model with a larger token input size if wanted.
>
> **Number of skills in codebook.** We chose the number of skills in the codebook and the horizon roughly based on our estimate of the nature of the dataset and the number of individual sub-tasks. Our ablation study in Appendix Section F1 shows that given a fixed horizon, changing the number of skills in the codebook does not affect performance much unless we increase them to a very large number - where the optimization problem becomes challenging to solve.
>
> **Figure Fixes.** We will improve the clarity of our figures and make them bigger in our next version.
>
>
> [1] Sutton et al - Between MDPs and semi-MDPs: A framework for temporal abstraction in reinforcement learning
>
> [2] Bacon et al - AAAI 2017 - The Option-Critic Architecture
>
> [3] Li et al - ICLR 2020 - Sub-policy Adaptation for Hierarchical Reinforcement Learning

---

> > ### Comment · Reviewer_qGvo · 2022-08-06
> > **Response**
> >
> > I thank the authors for answering my concerns.
> >
> > I agree that the skills learned are more controllable in LISA and I’m now confident that the proposed setup would indeed learn more controllable skills.
> >
> > Regarding sampling of skills, as the authors noted that sampling the skills after every H is an open area of research, especially in the Options literature, how was H chosen for the experiments shown in the paper? In the case when H is set to +inf, would the authors think that LISA would perform at a similar level / outperform DIAYN, as in this case, the difference would be sampling a more “known” skill instead of sampling randomly.

---

> > > ### Author Response · Authors · 2022-08-09
> > > **Response**
> > >
> > > We are delighted that our responses addressed your concerns and convinced you of the controllability strength of LISA. We answer the additional questions below:
> > >
> > > **Choice of H.** Great question! In our experiments in Table 1, we chose rough values for the horizon and the number of options without careful tuning, simply based on the nature of the dataset and our estimate of the number of skills required. These hyperparameter choices are by no means exhaustive nor optimal as our ablation study in appendix F1 suggests that our choice of horizon is perhaps sub-optimal (we used H=10 for our experiments in Table 1 but appendix section F1 suggests that H=5 is better). The ablation study also suggests that the performance is fairly stable for a reasonable range of hyperparameter choices.
> > >
> > > Regarding the comparison to DIAYN when H is set to +inf, this is not really a fair comparison since DIAYN is unsupervised and LISA works in an IL setting. If we had to guess, we’d expect our skills to be more useful purely because of the additional supervision and language conditioning.

---

### Meta-Review · Area_Chair_V1Tw · 2022-08-26

**Recommendation:** Accept
**Confidence:** Certain

**Metareview:**

This paper provides a hierarchical approach to imitation learning guided by language, which the reviewers unanimously found empirically compelling, as well as of potential interest to the broader NeurIPS community. I am happy to go with the consensus and recommend acceptance.

**Award:**

No

---

### Decision · Program_Chairs · 2022-09-14

Accept